eawe european academy of wind energy
Open Access WIND ENERGY SCIENCE DISCUSSIONS

# How wind speed shear and directional veer affect the power production of a megawatt-scale operational wind turbine

Patrick Murphy[1,2,3], Julie K. Lundquist[2,3], Paul Fleming[3]

[1]Department of Atmospheric Sciences, University of Washington, 408 ATG, Seattle, WA 98195-1640, United States
[2]Department of Atmospheric and Oceanic Sciences, University of Colorado Boulder, 20 UCB, Boulder, CO 80309, United States
[3]National Wind Technology Center, National Renewable Energy Laboratory, Golden, CO 80401, United States

*Correspondence to* Patrick Murphy (patmurph@uw.edu)

**Abstract.** Most megawatt-scale wind turbines align themselves into the wind as defined by the wind speed at or near the center of the rotor (hub height). However, both wind speed and wind direction can change with height across the area swept by the turbine blades. A turbine aligned to hub-height winds might experience suboptimal or superoptimal power production, depending on the changes in the vertical profile of wind, or shear. Using observed winds and power production over 6 months at a site in the high plains of North America, we quantify the sensitivity of a wind turbine's power production to wind speed shear and directional veer as well as atmospheric stability. We measure shear using metrics such as α (the log-law wind shear exponent), $\beta_{bulk}$ (a measure of bulk rotor-disk-layer veer), $\beta_{total}$ (a measure of total rotor-disk-layer veer) and rotor-equivalent wind speed (REWS), a measure of actual momentum encountered by the turbine by accounting for shear). We also consider the REWS with the inclusion of directional veer, $REWS_{\theta}$, although statistically significant differences in power production do not occur between REWS and $REWS_{\theta}$ at our site. When REWS differs from the hub-height wind speed (as measured either by the lidar or a transfer function-corrected nacelle anemometer), the turbine power generation also differs from the mean power curve in a statistically significant way. This change in power can be more than 70 kW, or up to 5% of the rated power for a single 1.5-MW utility-scale turbine. Over a theoretical 100-turbine wind farm, these changes could lead to instantaneous power prediction gains or losses equivalent to the addition or loss of multiple utility-scale turbines. At this site, REWS is the most useful metric for segregating the turbine's power curve into high and low cases of power production when compared to the other shear or stability metrics. Therefore, REWS enables improved forecasts of power production.

## 1 Introduction

Wind energy is already the second largest source of renewable energy in the United States and is the fastest growing source of renewable energy, providing 6.3% of the total energy in the United States (EIA, 2017). As wind energy continues to grow, so will the challenge of predicting power output and integrating that power with the rest of the electric grid (Marquis et al., 2011; Woodford, 2011; Xie et al., 2011; Vittal and Ayyanar, 2013; Heier, 2014; Heydarian-Forushani et al., 2014; Sarrias-Mena et al., 2014).

Currently, wind farm operators and control engineers rely on wind turbine power curves to predict the power production of a given model of turbine for various inflow wind speeds (Brower, 2012). The inflow wind speeds are typically measured by instrumentation on top of the nacelle at or near hub height, where the blades of a turbine connect to its hub. Wind turbines are designed to optimize these inflow wind speeds by orienting themselves into the inflow. Typical turbines use a wind vane located on top of the hub to determine the wind direction at that altitude. The turbine then rotates (yaws) into that inflow, so that the hub is aligned with and parallel to the wind vane (Fleming et al., 2014; Wan et al., 2015). This yaw correction happens periodically, and the exact frequency depends on the specific turbine and many other factors. However, hub-height wind speeds and directions do not necessarily represent the inflow across the turbine rotor disk. Wind speed and direction can change with height across the rotor disk, a phenomenon known as shear. “Wind shear” simply considers the change of wind speed with height, whereas a change in wind direction is considered “wind veer” (Holton, 1992). In atmospheric science, the direction of the change of wind direction can also be useful; in the Northern Hemisphere, clockwise rotation with height is considered “veering” while counterclockwise rotation is considered “backing.”

Several common atmospheric phenomena cause vertical wind shear or veering/backing over the depth of a turbine’s rotor disk. Wind speeds tend to increase with height in the atmosphere as the effects of surface friction decrease. In the planetary boundary layer this increase is, on average, logarithmic (Tennekes, 1973). Flows over land exhibit more shear because friction is larger over the land than the ocean. At night, the lack of mixing from convective eddies allows winds in the boundary layer to decouple from the surface, such that both wind speed and direction can change with height (Blackadar, 1957; Walter et al., 2009). Nocturnal low-level jets, characterized by a maximum in wind speed in the stable boundary layer, often form over the Great Plains because of the decoupling phenomenon and inertial oscillations as well as the nocturnal change of the thermal wind (Blackadar, 1957; Whiteman et al., 1997; Banta et al., 2002; Vanderwende et al., 2015). Shear or veer associated with inertial oscillations also occur because of frontal passages (Lundquist, 2003). Low-level jets can form offshore, leading to wind speed shear (Kraus et al., 1985; Hsu, 1988; Smedman et al., 1993; Ranjha et al., 2013; Pichugina et al., 2017) or wind directional veer (Bodini et al., 2019b) across the altitudes of a turbine rotor disk. Turbines located near the mouth of a canyon might experience shear effects of nocturnal valley exit jets (Banta et al., 1996; Jiménez et al., 2019). Warm and cold air advection can lead to directional veer (Holton, 1992). Outflow from thunderstorms can introduce density

Open Access WIND ENERGY SCIENCE DISCUSSIONS

currents that affect both speed shear and directional veer (Goff, 1976; Lynch and Cassano, 2006). Finally, land-based topographic effects allow for the formation of localized circulations and microclimatic effects that could interact with the mean airflow across a rotor disk and create shear (Mahrt et al., 2014; Fernando et al., 2019).

Over the past three decades, shear and turbine power production have been related by various observational studies. In 1990, shear affected power curves, as seen in observations of three 2.5-MW turbines (Elliott and Cadogan, 1990). Shear decreases the power coefficient, compared to nonshear cases, for multimegawatt turbines (Albers et al., 2007). Diurnal variations of power production have been found resulting from diurnal variations of shear in a region of complex terrain at a site in the interior of the continental United States (Antoniou et al., 2009). Increases in power of a theoretical wind farm using observational shear values (rather than no-shear values) could be up to 0.5%, while decreases in power could approach 3% as found by Walter et al. (2009). Model power curves (or power surfaces where the power production of a turbine is a function of both wind speed and air density) made from equivalent wind speeds from actual 2.5-MW turbine power data are more accurate than a standard power curve (Vahidzadeh and Markfort, 2019).

In addition, other simulation-based studies quantify the magnitude of the effects found observationally (Pedersen, 2004; Wagner et al., 2010). The power productions found in both Pedersen (2004) and Wagner et al. (2010) are dependent on the magnitude of the shear and whether the shear is based on direction or velocity. Wagner et al. (2010) additionally find that directional veer was less influential on the power production than speed shear. Sanchez Gomez and Lundquist (2019) suggest a combination of directional veer and shear should be considered.

Actual observations of wind shear and veer exhibit a significant variety of shapes (Pé et al., 2018), as shown in Figure 1, with four wind speed profiles from vertically profiling Doppler lidar and relevant idealized linear and logarithmic profiles. All profiles show differences between the idealized profiles and the actual profiles and differences between the 80-m wind speed (effectively the height of the nacelle anemometer and vane) and the speeds at other heights. Though the first three of the four real profiles (Fig. 1a,b,c) appear similar to the idealized profiles, differences occur between the winds at all non-80-m heights and the idealized profiles (Fig. 1e,f,g). The winds at 80 m (effectively the height of the nacelle anemometer and vane) clearly differ from the winds at other heights as well. The fourth profile (Fig. 1d) shows the most nonlinear and nonlogarithmic wind speed profile and also shows the greatest difference between the 80-m wind speeds and wind speeds at other heights (Fig. 1h). Because the differences exist between the height levels for all profiles, the 80-m wind speed and thus the nacelle wind speed are not truly representative of the average wind speed across the rotor for any of the wind speed profiles.

eawe european academy of wind energy
Open Access WIND ENERGY SCIENCE DISCUSSIONS

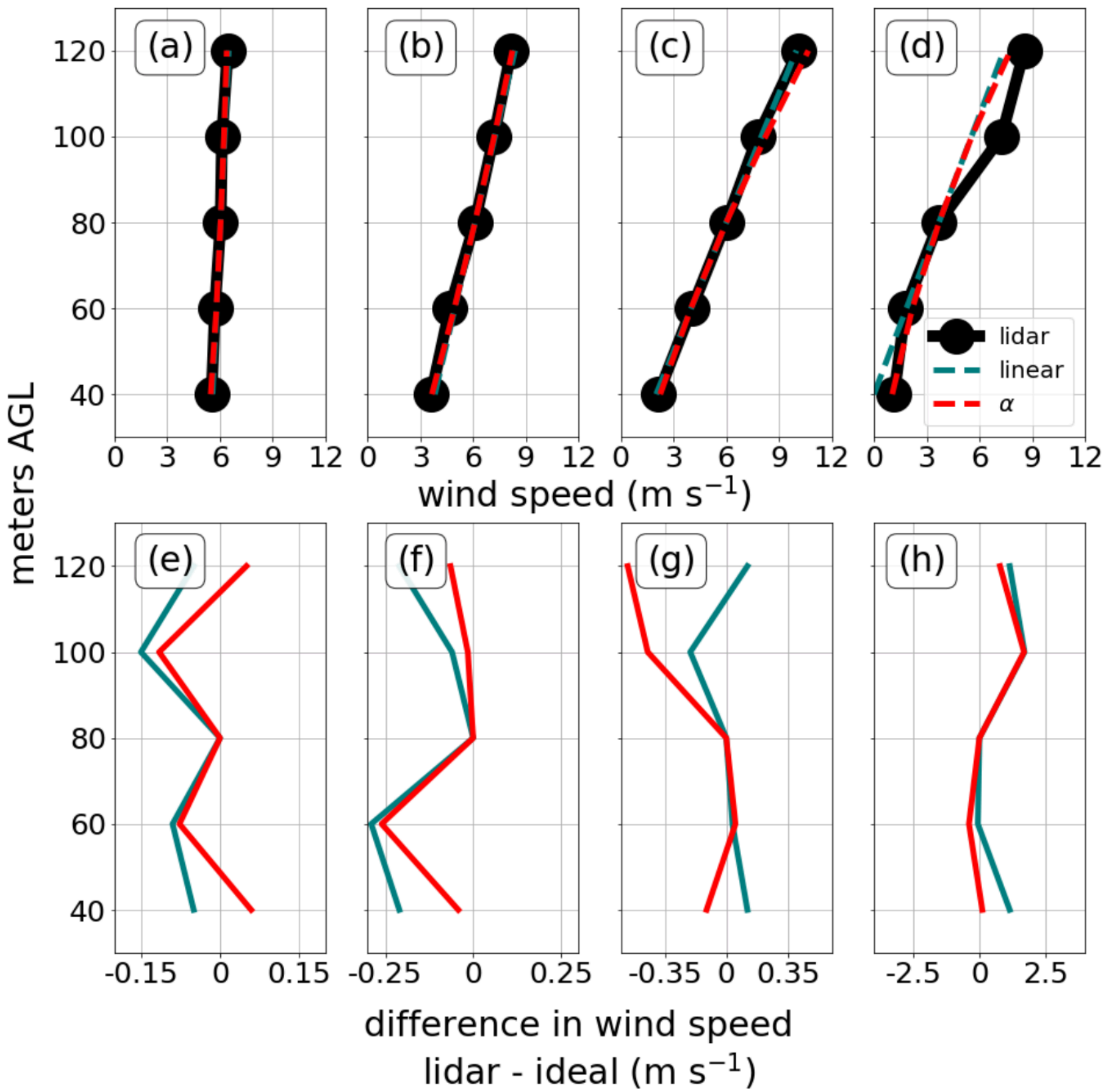

**Figure 1: Top row: Four wind speed profiles as measured by the lidar (black line with circle markers), with measurement heights above ground level (AGL) denoted by circles. Teal dashed lines denote the linear profile fit to the real profile; red dashed lines denote the power law profile fit to the real profile. The $\alpha$ values in the top row calculated between 40 m and 120 m above ground level (AGL) are (a) 0.14, (b) 0.74, (c) 1.42, and (d) 1.83. Bottom row: The difference (m s$^{-1}$) between the lidar wind speed and the idealized linear (teal) and logarithmic (red) profiles.**

This poor representation has consequences for turbine power production. The power produced by a turbine varies with the cube of the inflow wind speed in region II of a power curve (where turbines spend most of their time operating and where each of the profiles were taken from) as seen by:

$$P(t) = \frac{1}{2}\rho A C_p U(t)^3, \quad (1)$$

where $P(t)$ is the power at a given time $t$, $\rho$ represents the air density, $A$ represents the area swept out by the rotor disk, $C_p$ represents the coefficient of power which has a maximum of 0.59, and $U(t)$ represents the inflow wind speed across the rotor disk at time $t$ (Brower, 2012). Directional veer can mitigate or worsen the effects of speed shear.

A rotor-equivalent wind speed (REWS) metric can describe the actual momentum encountered by a turbine rotor disk by accounting for the vertical shear. The simplest REWS, proposed by Wagner et al. (2008), accounts for only the wind speed shear and does so by dividing a turbine's rotor disk into discrete vertical layers or bins:

$$REWS_{Wagner} = \sqrt[3]{\frac{1}{A}\left(\sum_i \bar{u}_i^3 A_i\right)}, \quad (2)$$

where $REWS_{Wagner}$ is the equivalent wind speed, $A$ represents the area swept out by the rotor disk, $A_i$ represents the area of a discretized section of the rotor disk, and $u_i$ represents the wind speed measured for the given section. Using a blade element momentum model to simulate a 3.6-MW turbine, Wagner et al. (2008) show that power production correlates better with the REWS than with the hub-height wind speed. Later work specified a $\mathrm{REWS}_\theta$, which considers both speed shear and directional veer (Wagner et al., 2010; Choukulkar et al., 2015; Clack et al., 2016). Though similar to $REWS_{Wagner}$, this method considers only the orthogonal component of the inflow wind speed to the plane of the turbine's rotor disk at each height bin. Although the combined effects of wind speed shear and wind directional veer on a turbine's power production are often stronger than either speed shear or directional veer alone, speed shear exerts more influence than directional veer in most circumstances. Turbulence can also affect the momentum accessible to a wind turbine rotor and is accounted for in the method of Choukulkar et al. (2015).

Although former studies used REWS and similar metrics to explore the impact of shear and atmospheric stability on the prediction of power production from megawatt-scale turbines (Elliott and Cadogan, 1990; Rohatgi and Barbezier, 1999; Pedersen, 2004; Sumner and Masson, 2006; Albers et al., 2007; Van den Berg, 2008; Antoniou et al., 2009; Walter et al., 2009; Belu and Koracin, 2012; Wharton and Lundquist, 2012b; Vanderwende and Lundquist, 2012; Sanchez Gomez and Lundquist, 2019; Vahidzadeh and Markfort, 2019), a more recent study (Sark et al. 2019) concludes that turbines in regions with flat terrain do not benefit from using REWS rather than a hub-height wind speed. Here, we explore how different regimes of speed and directional veer across the turbine rotor disk affect power production of a megawatt-scale onshore turbine in a wind farm in the high plains of North America. Defining several wind speed and direction-based shear metrics,

we compare power production in different regimes. We distinguish the importance of wind shear and veer and suggest the influence of topography. Finally, we address how the regimes differ from a mean power curve.

In Section 2, we describe the observational data set and data processing steps. In Section 3, we define REWS metrics and other shear metrics to characterize speed shear and directional veer. In Section 4, we describe distributions of the metrics for this site, demonstrate the superiority of REWS over hub-height wind speed for power prediction, and explore how other shear metrics relate to power production. We summarize results in Section 5 and pose suggestions for future work.

## 2 Observational data set

The data discussed in this paper were collected as part of a wake steering campaign conducted by the National Renewable Energy Laboratory on five turbines at a commercial wind farm in the high plains of North America (Fig. 2, more details in Fleming et al., 2019). Data for this study were collected from 04:00 UTC on May 2, 2018, through 23:59 UTC on October 31, 2018. This paper focuses on the turbine shown in red in Fig. 2. Although this turbine is not waked under typical wind directions at the site, waked data are removed as described in Section 2.3. Wind profile observations are collected by the lidar 350 m east-northeast of the chosen turbine.

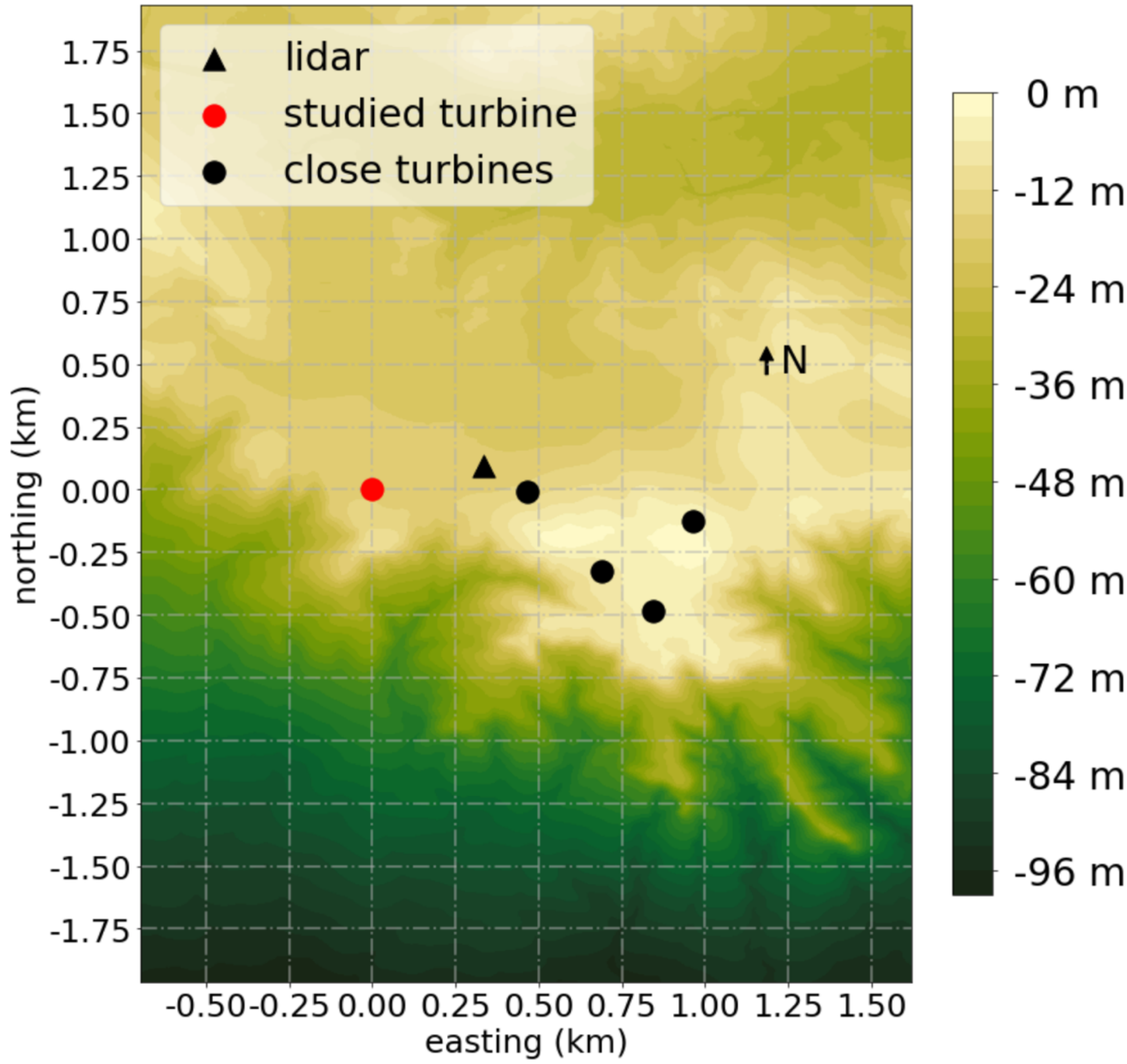

**Figure 2: Layout of relevant equipment. The negative elevation measurements represent meters below the maximum elevation in the figure. Exact locations and elevations are not given at the request of the wind farm owner/operator. The westernmost red circle represents the turbine studied in the paper. The triangle represents the vertically profiling Doppler lidar and meteorological tower (co-located). The four black circles to the east-southeast represent other turbines that could potentially wake the lidar and studied turbine.**

## 2.1 Turbine data set

The turbine and lidar are located at the same elevation on a flat plateau. To the east and southeast, four other turbines are located within 1 km (Fig. 2). Methods for filtering waked data are described in Section 2.3. The plateau's escarpment, which descends around 100 m per kilometer, lies south of the focus area. Southerly winds are not filtered out of the data set because such terrain can lead to the formation of speed shear and directional veer. The northerly fetch is relatively complex as well, though to a much lesser extent than the southerly fetch. To the northeast, the terrain descends to a depth about half that of the escarpment to the south and does so over a much gentler slope. To the northwest, the terrain descends to a depth about one-ninth that of the south. To the north the terrain descends to a depth around one-fourth that of the south.

The turbine of interest is a 1.5-MW General Electric super-long extended cold weather extreme model with a cut-in wind speed of 3.5 m s$^{-1}$, a rated wind speed of 14 m s$^{-1}$, and a cut-out speed of 25 m s$^{-1}$. Both the turbine rotor diameter $D$ and the hub height are nominally 80 m. Power production, nacelle wind speed and direction, fault codes (such as "turbine ok," "weather conditions," "grid loss," and so on), and blade pitch angle from the turbine were recorded at 1 Hz by the turbine's supervisory control and data acquisition (SCADA) systems. Data processing methods applied to the data set regarding the turbine data and potential curtailments and periods of inactivity are addressed in Section 2.3.

For comparison to the power production, we consider the power curve of a generic 1.5-MW turbine (Schmitz, 2015).

### 2.2 Lidar data set

Wind speed and direction profiles are collected by a Leosphere WINDCUBE v2 located ~ 4 $D$ east-northeast of the turbine, identical to the model used in Bodini et al. (2019a) and Bodini et al. (2019b). The lidar takes three-dimensional wind speed and direction measurements at approximately 1 Hz every 20 m above ground level from 40 m to 180 m. The lidar samples sequential line-of-sight velocity measurements along the four cardinal directions at 28º ($\theta$) from the vertical, followed by an additional beam oriented vertically. It completes a cycle of measurements nearly every five seconds. The lidar synthesizes the beams' line-of-sight measurements into a 1-Hz sample of horizontal and vertical wind speed component measurements. The manufacturer reports horizontal wind speed accuracies of 0.1 m s$^{-1}$ and wind direction accuracies of 2º. Time lags between the lidar and the turbine were not considered because of challenges in considering the advection of the wind. The horizontal wind speed components, $u$ (west-east) and $v$ (south-north), are found by:

$$u = \frac{V_{los,E} - V_{los,W}}{2 \sin \theta}, \tag{3}$$

$$v = \frac{V_{los,N} - V_{los,S}}{2 \sin \theta}, \tag{4}$$

where $V_{los}$ denotes the line-of-sight velocities at the cardinal directions north (N), east (E), south (S), and west (W).

A meteorological tower with a Campbell cSAT3 sonic anemometer at 10 m, a Vaisala PTB110 pressure sensor at 1.5 m, a relative humidity measurement at 2 m, and an RTD temperature measurement at 2 m is co-located with the lidar. To quantify atmospheric stability, the Obukhov length $L$ is calculated using 20-Hz 10-m sonic anemometer data, 1-Hz 1.5-m pressure data, 1-Hz 2-m temperature measurements, and 1-Hz 2-m relative humidity measurements:

$$L = \frac{-u_*^3 \overline{\theta_v}}{k g \overline{w' \theta_v'}}, \tag{5}$$

where $k = 0.4$ is the von Kármán constant, $g$ is the acceleration of gravity 9.81 m s$^{-2}$, $u_*$ is the friction velocity calculated by: $u_* = [\overline{u'w'}^2 + \overline{v'w'}^2]^{1/4}$, $\theta_v$ in the numerator is the virtual potential temperature in Kelvin calculated from the 1-Hz 2-m temperature $T_d$ in Celsius with modifications from the 1-Hz 2-m relative humidity $RH$ and 1.5-m pressure $p$ to convert the temperature to virtual temperature by:

$$e_s = 6.11x10^{\frac{7.5T_d}{237.3+T_d}}, \quad (6)$$

$$w = \frac{RH}{100} 621.97 \frac{e_s}{p-e_s}, \quad (7)$$

$$T_v = T_d \frac{(1+w/.622)}{(1+w)}, \quad (8)$$

Further modifications from the 1.5-m pressure $p$ by: $\theta_v = (T_v + 273.15) * \left(\frac{p_0}{p}\right)^{R/c_p}$ with $p_0 = 1000$ mb and $R/c_p \approx .286$ convert the virtual temperature to a virtual potential temperature; $\theta_v$ in the denominator is the virtual potential temperature in Kelvin calculated from the 20-Hz virtual temperature from the speed of sound and the same potential pressure calculation as the numerator, and $w'\theta_v'$ is the kinematic sensible heat flux. The covariances for the heat flux and friction velocity are calculated from a Reynolds decomposition over a 30-minute averaging time.

To quantify atmospheric stability we use two regimes, convective and stable, based on the nondimensional stability parameter (otherwise known as the surface-layer scaling parameter). $\zeta = z/L$ is used, where $z$ is the height above ground level (10 m) of the flux measurements for $L$. Note that these categories are similar but not identical to the stable and nonstable categories of Fleming et al. (2019). Convective conditions occur during $0 < \zeta < \infty$, while stable conditions occur when $-\infty < \zeta < 0$. Values further from 0 are stronger stabilities. Values that could be considered neutral ($-0.01 \leq \zeta \leq 0.01$ as in Wharton and Lundquist, 2012a) only occur in 3.9% of the postfiltered data, and so are classified as stable or convective based on their sign.

Figure 3 shows the dominant winds as measured by the lidar at 80 m above ground level (hub height) during the campaign through three wind roses using (a) all data, (b) convective stability data, and (c) stable stability data. This figure is made with prefiltered data.

WIND ENERGY SCIENCE
DISCUSSIONS

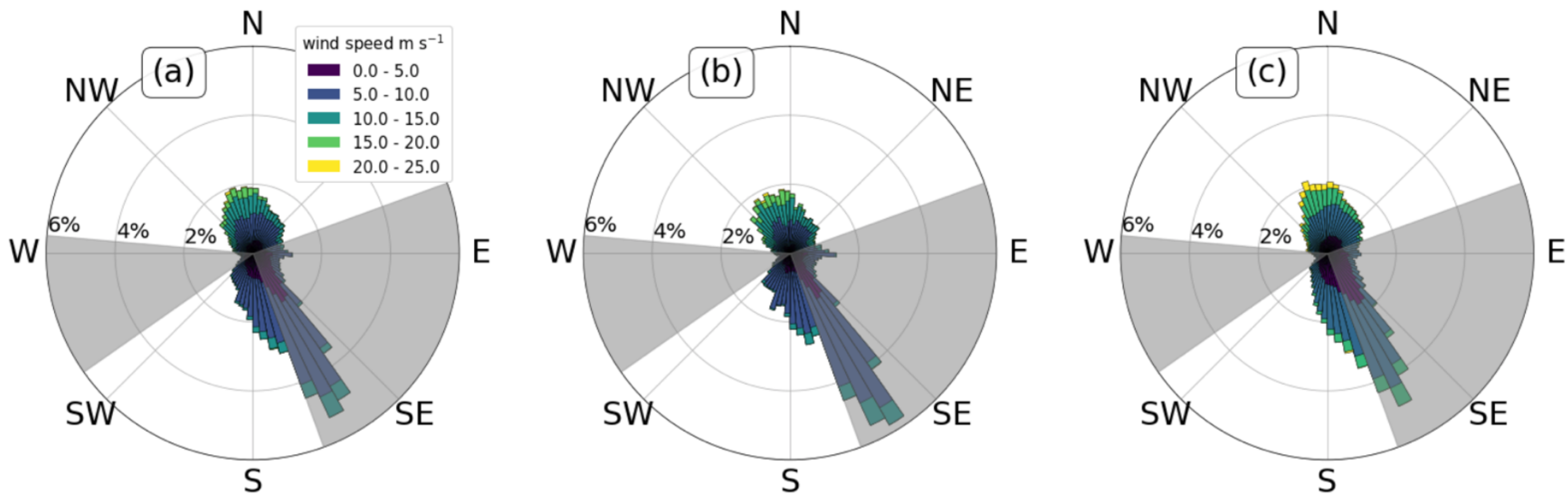

**Figure 3: Lidar winds at 80 m above ground level from 04:00 UTC on May 2, 2018, until 23:59 on October 31, 2018; (a) all stabilities, (b) convective stabilities, and (c) stable stabilities. Data within the grey areas are later rejected because of possible wake effects, as detailed in Section 2.3.**

## 2.3 Data filtering

Data collection extended from 04:00 UTC on May 2, 2018, until 23:59 on October 31, 2018, nearly 15.8x10$^6$ seconds (nearly 6 months of data). Several data filters are applied.

Because of our focus on power production, we first removed time periods with turbine fault codes given in the SCADA data. Data are considered acceptable for four SCADA codes, "turbine ok," "turbine with grid connection," "run up/idling," and "weather conditions." The codes that are filtered out are related to maintenance, repair, grid loss, stops, wind direction curtailments, and further codes that are determined by the utility company to be bad but are not specified further. This filter removed 14.2% of the data.

A further 11.5% of the data was removed because of the turbine not producing power (power greater than 0 kW). Another 8.4% of the data was removed because of the lidar not functioning on at least one of its five measurement heights within the turbine rotor disk.

Blade pitch angles greater than 6º were filtered out as well to remove data that could be affected by curtailments. Blade pitch angles were used to filter data in other studies (St. Martin et al., 2016; Sanchez Gomez and Lundquist, 2019). We discarded data with blade pitch angles exceeding 6º for this 1-Hz data set. This threshold was chosen experimentally to retain as much data as possible while still removing outliers. This approach removed a further 8.1% of the data.

Times when ζ could not be calculated because of issues with any of the instrumentation used in creating ζ were removed. This filter removed around 0.67% of the data.

Because of our focus on power production in region II of the turbine, we only considered data with REWS less than or equal to the turbine's rated wind speed. Once the REWS is at rated speed, the turbine can be assumed to be operating at rated power, regardless of whether the REWS is greater or less than the nacelle wind speed. This filter removed 0.48% of the data.

Once the data had been filtered, we considered turbine yaw error. The lidar 80-m wind direction may differ from the turbine nacelle wind vane (Fig. 4a). Differences in direction greater than 25º in either direction were filtered out because of the large effects of yaw misalignment, as shown in Fig. 4b, which shows the theoretical effect of the cosine, cosine$^2$, and cosine$^3$ relationships between the yaw misalignment and power production by a yaw-misaligned turbine (Pedersen, 2004; Choukulkar et al., 2015; Mittelmeier and Kühn, 2018). The curve that a yaw-misaligned turbine follows depends on the aero-elastic properties of a given turbine itself (Fleming et al., 2014). Note that although these theoretical power impacts are symmetric, some work (Wagner et al., 2010; Sanchez Gomez and Lundquist, 2019) suggests that veering and backing have nonsymmetric effects. This filter removed 4.1% of the data.

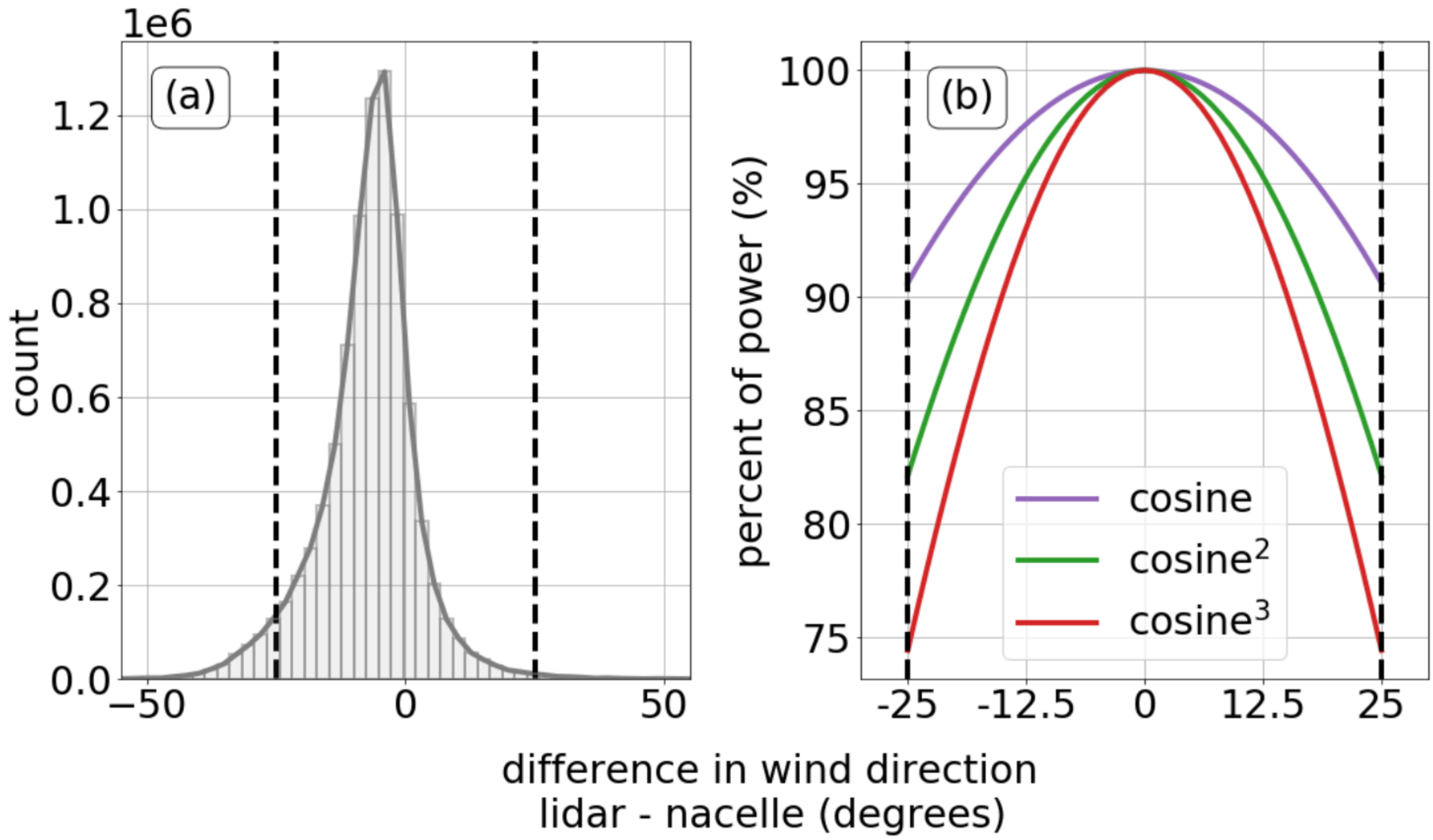

**Figure 4: (a) Histogram of the occurrences of yaw misalignments (differences in wind direction between the 80-m lidar measured wind and the nacelle hub-height measured wind). Vertical black dashed lines denote -25º and 25º, which are limits imposed by the authors such that larger and smaller misalignments are filtered out. (b) Different theoretical effects of the yaw misalignment on power production for a misaligned turbine following different proposed cosine curves.**

Finally, wind directions were removed during which either the lidar or the turbine could be waked (grey areas in Fig. 3 resulting from the turbine locations shown in Fig. 1). To specify these directions, the difference in wind speed between the lidar at 80 m (hub height) and nacelle is calculated for 1º direction bins (direction measured by the lidar). A 99% two-tailed confidence interval is calculated for each bin:

$$\bar{x} - t_{.005}\frac{\sigma}{\sqrt{N-1}} \leq \mu_{metric} \leq \bar{x} + t_{.005}\frac{\sigma}{\sqrt{N-1}}, \quad (9)$$

where $\mu_{metric}$ is the true population mean of the wind speed difference in a bin, $\bar{x}$ is the sample mean of the wind speed difference in a bin, $t_{.005}$ is the critical value of $t$ at 99% confidence, $\sigma$ is the sample standard deviation of the wind speed difference in a bin, and $N$ is the number of values in the bin (Fig. 5) (Wilks, 1962). Based on Fig. 5, we removed directions where the 99% confidence interval on the mean difference between the two wind speeds over a 15º group of direction bins changed smoothly to be 1 m s$^{-1}$ different from the mean without inclusion of those directions (70º–160º and 235º–275º). This removal was done iteratively by hand by changing the removed directions (and thus changing the mean without those

directions). The southeasterly flow does not completely conform to the quantitative process because of the physically based inflection point, where the turbine is waked stronger closer to the east and the lidar is waked stronger closer to the south because of the layout of the equipment (Fig. 2). However, those directions were removed as well. Discarding these wind directions removed an additional 22.2% of the data. We repeated the same process based on the nacelle wind direction, which resulted in smaller ranges of wind directions (not shown). The wider direction bins (from the lidar direction) were filtered.

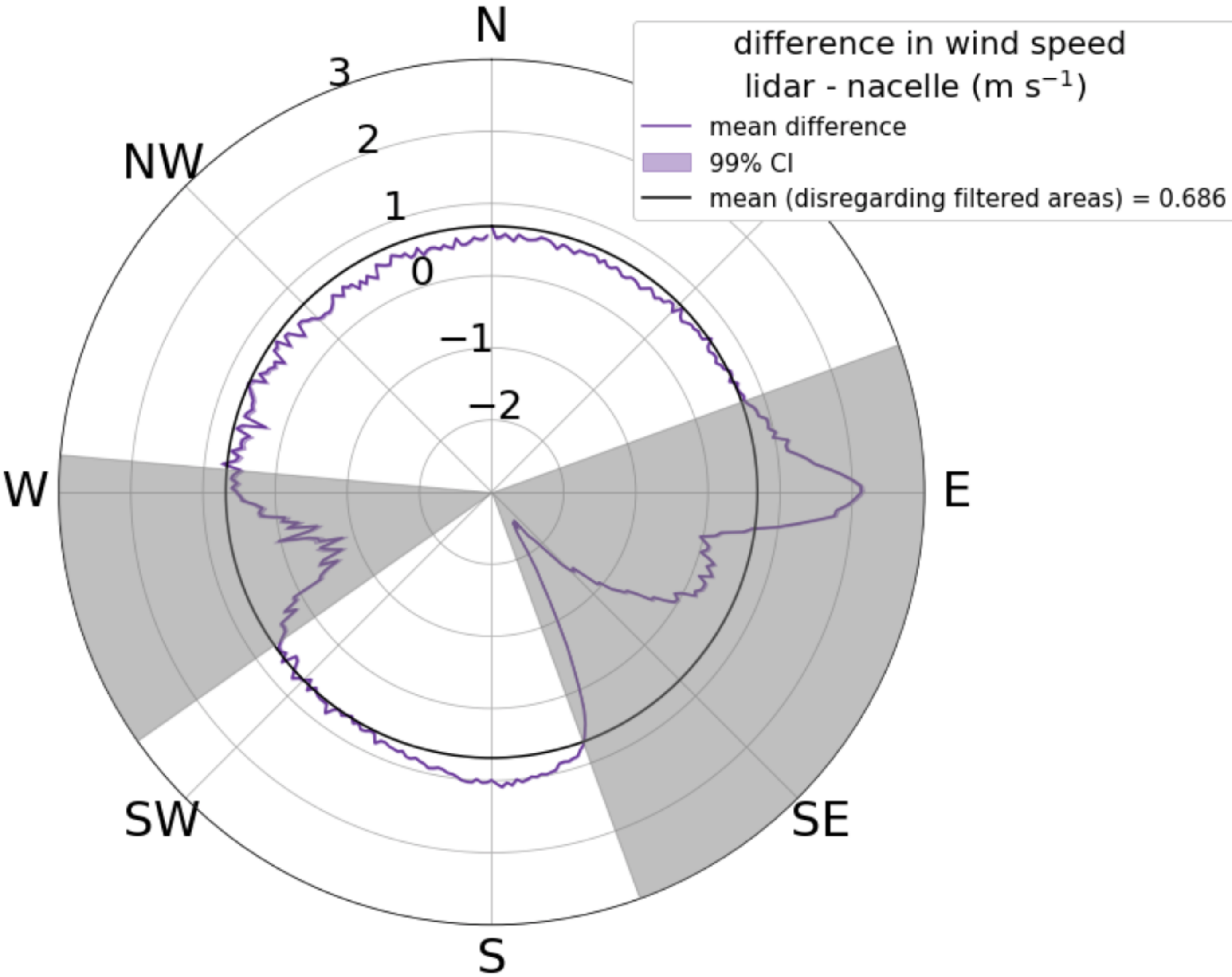

**Figure 5: 99% confidence intervals on the difference in wind speed between lidar 80-m wind speed and hub-height nacelle wind speed binned by 1° direction bins. Data within the grey areas were rejected because of possible wake effects (Section 2.3).**

All of these filtering processes left a total of nearly 4.8x10$^6$ seconds for analysis, or the equivalent of almost 2 months of 1-Hz data (30.4% of the total). Subsequent analyses were applied to this subset of the data. All subsequent data percentage plots are based on the filtered data set.

## 3 Methods

Calculations of shear metrics are described in Section 3.1. Methods for creating power curves are described in Section 3.2.

### 3.1 Shear calculation methods

REWS represents the effect of wind speed shear across the rotor disk using discretized wind speed profiles. REWS is calculated by:

$$\text{REWS} = \sqrt[3]{\frac{\sum_{i=0}^{i=3} A_{z(i)\ to\ z(i+1)}(\frac{U_{z(i+1)}+U_{z(i)}}{2})^3}{A}}, \quad (10)$$

where $z$ represents a height from the list of discrete heights that the lidar measures across the rotor diameter (40 m, 60 m, 80 m, 100 m, and 120 m) and $\{i \mid (0,1,2,3,4)\}$ indexes through those heights, $A_{z(i)\ to\ z(i+1)}$, represents the area of the rotor disk between two discrete heights $z(i)$ and $z(i+1)$, $U_{z(i)}$ represents the wind speed at the height $z(i)$ and $U_{z(i+1)}$ represents the wind speed at the height z$(i+1)$, and $A_{turbine}$ represents the overall area of the turbine rotor disk (approximated to be a perfect circle of radius 40 m for our purposes). This calculation follows Wagner et al.'s (2008) method, but with slight modifications because of the lidar data collection at discrete heights, including the rotor disk bottom and top, rather than heights found in the middle of each discrete interval (Fig. 6). This averaging assumes that the winds vary linearly across each 20-m span of the turbine and that their average represents the true inflow across that area.

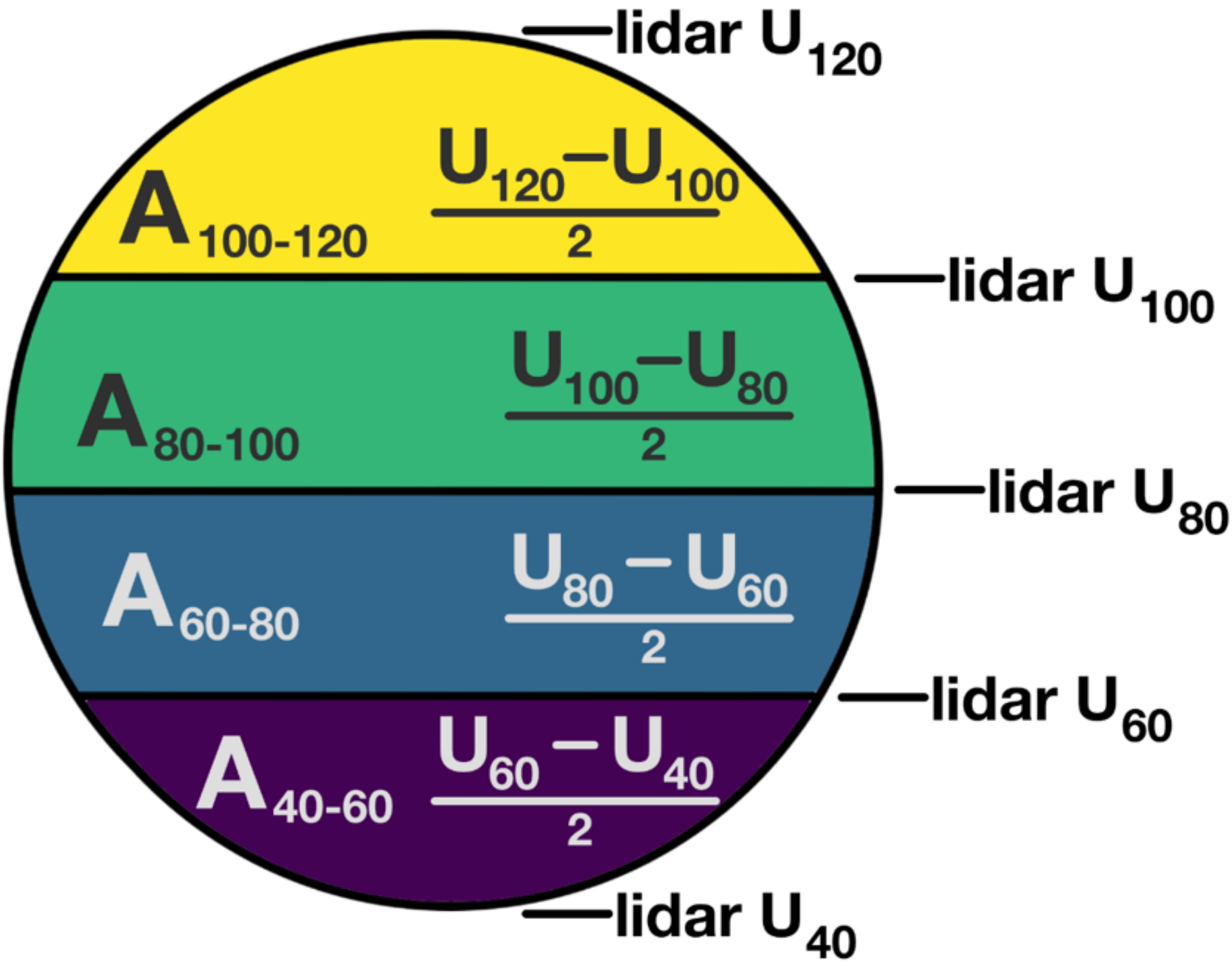

**Figure 6: Schematic for calculation of the REWS. The turbine rotor disk (circle) is divided into four discrete areas. $A_i$ denotes the area of the colored section from z(i) to z(i+1). The $\frac{U_{z(i+1)}+U_{z(i)}}{2}$ terms denote the averaged horizontal wind speed used for a given colored area. Lidar measurement heights are shown at right.**

We use the REWS to calculate a difference from the nacelle wind speed as $\Delta\text{REWS}_{N-NTF}$:

$$\Delta\text{REWS}_{N-NTF} = \text{REWS} - U_{nacelle} - NTF, \quad (11)$$

where REWS is as calculated in Eq. 11 and $U_{nacelle}$ is the wind speed measured by the nacelle mounted anemometer, and $NTF = \overline{(U_{lidar} - U_{nacelle})}$ is a simple nacelle transfer function (NTF). This simple NTF is a bias calculation between the lidar wind speed and nacelle wind speed of 0.686 m $s^{-1}$ based on all wind directions over the entire filtered data set.

Although the $NTF$ varies slightly with direction (Fig. 5), those variations are less than 10% of the $NTF$ itself. A true NTF is not applied in part because the lidar does what a true transfer-function-corrected nacelle measurement is supposed to do: measure the wind speed most accurately, disregarding rotor wake effects. The application of the $NTF$ shifts the peak of the histogram of $\Delta\text{REWS}_{N-NTF}$ to 0 as well (Fig. 8a).

A similar metric comparing the lidar hub-height wind speed with the REWS, $\Delta\text{REWS}_L$, is calculated by:

$$\Delta\text{REWS}_L = \text{REWS} - U_{lidar}, \tag{12}$$

where $U_{lidar}$ is the hub-height lidar wind speed measurement.

$\Delta\text{REWS}_{N-NTF}$ and $\Delta\text{REWS}_L$ quantify whether using the nacelle wind speed underestimates ($\Delta\text{REWS}_{N-NTF}$ or $\Delta\text{REWS}_L$ are negative) or overestimates ($\Delta\text{REWS}_{N-NTF}$ or $\Delta\text{REWS}_L$ are positive) the rotor-disk-integrated winds encountered by the turbine.

The REWS with direction, $\text{REWS}_\theta$, represents the effect of both wind speed shear and wind directional veer across the rotor disk using discretized wind speed and direction profiles (Choukulkar et al., 2015). Similar to how Eq. 11 integrates wind speed across the rotor disk, $\text{REWS}_\theta$ integrates the normal component of the flow across the rotor disk and therefore considers the directional veering and backing:

$$\text{REWS}_\theta = \sqrt[3]{\frac{\sum_{i=0}^{3} A_{z(i)\ to\ z(i+1)} \left(\frac{U_{z(i+1)}\cos\left(\Delta\theta_{z(i+1)}\right) + U_{z(i)}\cos\left(\Delta\theta_{z(i)}\right)}{2}\right)^3}{A}}, \tag{13}$$

where $z$, $i$, $A_{z(i)\ to\ z(i+1)}$, $U_{z(i)}$, $U_{z(i+1)}$, and $A_{turbine}$ are as described for Eq. 11, and $\Delta\theta_{z(i)} = \theta_{lidar,z(i)} - \theta_{nacelle}$ is the difference between the lidar wind direction at height $z(i)$ and nacelle wind direction (and is always between -180 and 180 degrees). $\Delta\theta_{z(i)} < 0$ specifies that the lidar-measured wind direction is "to the left" of the turbine as seen facing upwind, while $\Delta\theta_{z(i)} > 0$ specifies the lidar wind direction is "to the right" of the turbine as seen facing upwind.

To quantify difference, $\Delta\text{REWS}_{\theta,N-NTF}$ is calculated by:

$$\Delta\text{REWS}_{\theta,N-NTF} = \text{REWS}_\theta - U_{nacelle} - NTF, \tag{14}$$

where $\text{REWS}_\theta$ is as calculated in Eq. 13, $U_{nacelle}$ is the wind speed measured by the nacelle-mounted anemometer, and $NTF$ is the simple nacelle transfer function discussed previously. The application of the $NTF$ also shifts the peak of the histogram of $\Delta\text{REWS}_{\theta,N-NTF}$ to 0.

Similarly, $\Delta\text{REWS}_{\theta,L}$ is calculated by:

$$\Delta\text{REWS}_{\theta,L} = REWS_{\theta} - U_{lidar}, \tag{15}$$

where $U_{lidar}$ is the hub-height wind speed measurement.

$\Delta\text{REWS}_{\theta,N-NTF}$ and $\Delta\text{REWS}_{\theta,L}$ quantitatively show whether using the nacelle wind speed underestimates ($\Delta\text{REWS}_{\theta,N-NTF}$ or $\Delta\text{REWS}_{\theta,L}$ are negative) or overestimates ($\Delta\text{REWS}_{\theta,N-NTF}$ or $\Delta\text{REWS}_{\theta,L}$ are positive) the rotor-disk-integrated winds encountered by the turbine, considering veering/backing.

Wind shear is also quantified with the wind shear exponent, $\alpha$ (Peterson and Hennessey, 1978; Emeis, 2013), calculated in a bulk fashion by considering only wind speed at the top and bottom of a vertical layer of atmosphere, presuming a logarithmically increasing profile:

$$\alpha = \frac{\log\left(\frac{U_{top}}{U_{bottom}}\right)}{\log\left(\frac{z_{top}}{z_{bottom}}\right)} \tag{16}$$

where $U_{top}$ and $U_{bottom}$ are the lidar-measured horizontal wind speeds at the top (120 m) and bottom (40 m) of the rotor disk, and $z_{top}$ and $z_{bottom}$ are the heights of 120 m and 40 m, respectively. While $\alpha$ may be simple to calculate and is thus widely used (Peterson and Hennessey, 1978; Wharton and Lundquist, 2012b; Vanderwende and Lundquist, 2012; Emeis, 2013), wind profiles may differ from a logarithmic profile across the rotor diameter of a turbine (Wagner et al., 2008). Additionally, $\alpha$ does not consider veering or backing, or even the magnitude of the wind speed.

We consider directional veer with two further metrics. The simplest metric, $\beta_{bulk}$, considers only differences of wind direction at the top and bottom of the rotor disk:

$$\beta_{bulk} = \frac{\theta_{top} - \theta_{bottom}}{z_{top} - z_{bottom}} \tag{17}$$

where $\theta_{top}$ and $\theta_{bottom}$ are the lidar-measured horizontal wind directions at the top (120 m) and bottom (40 m) of the rotor disk (values constrained to lie between -180 and 180), and $z_{top}$ and $z_{bottom}$ are the heights of 120 m and 40 m, respectively. $\beta_{bulk}$ resembles depictions of layer-wise directional veer in hodographs (MacKay, 1971), where the shear is only considered as a bulk quantity. A negative $\beta_{bulk}$ implies backing of the wind across the turbine rotor disk (the wind rotates counterclockwise as it increases in height), while a positive $\beta_{bulk}$ implies veer (the wind rotates clockwise as it increases in height). In a simulation, Wagner et al. (2010) found that a clockwise veer increases turbine power production while counterclockwise backing decreases the power produced because of differences in angle of attack for the turbine blades. However, Sanchez Gomez, and Lundquist (2019) found different results during an observational study such that veer leads

to a larger decrease on turbine power production than backing. The $\beta_{bulk}$ calculation does not consider any general yaw misalignment from the 80-m hub-height wind speed as measured by the lidar that might occur at the same time as directional shear. Thus, it is impossible to know whether power changes in $\beta_{bulk}$ conditions are a result of yaw misalignments or directional shear. Like $\alpha$, $\beta_{bulk}$ does not consider the hub-height wind speed.

A more discrete veer metric, $\beta_{total}$, considers shear at each level:

$$\beta_{total} = \frac{\sum_{i=0}^{3} |\theta_{z(i+1)} - \theta_{z(i)}|}{z_{top} - z_{bottom}} \quad (18)$$

where $z$ and $i$ are as described for Eq. 11, $\theta_{top}$ and $\theta_{bottom}$ are the lidar-measured horizontal wind directions at the top (120 m) and bottom (40 m) of the rotor disk, and $\theta_{z(i+1)} - \theta_{z(i)}$ is the difference between the lidar wind direction at height z($i$ + 1) and the lidar wind direction at height z($i$), constrained to be between -180 and 180 degrees. This measurement assumes that both veer and backing will decrease the power output of a turbine and will do so symmetrically. $\beta_{total}$ should be considered for cases where the directional veer is nonmonotonic across the rotor. Like $\beta_{bulk}$, $\beta_{total}$ does not consider yaw misalignment or the hub-height wind speed.

These metrics were visualized using an example lidar-measured wind profile (Fig. 7) during a time period with a $\zeta$ of 0.45 (convective). The turbine was producing power at this time, though the exact power is not given at request of the utility company. The nacelle wind speed was 4.50 m s$^{-1}$, the turbine was oriented to winds from 285º, the lidar wind speed at hub height was 3.7 m s$^{-1}$, and the lidar wind direction at hub height was 286.8º. The shear metrics vary: the REWS was 5.36 m s$^{-1}$, so the $\Delta$REWS$_{N-NTF}$ was 0.15 m s$^{-1}$ and the $\Delta$REWS$_{L}$ was 1.66 m s$^{-1}$; the REWS$_{\theta}$ was 5.28 m s$^{-1}$ with a $\Delta$REWS$_{\theta,N-NTF}$ of 0.08 m s$^{-1}$ and $\Delta$REWS$_{\theta,L}$ of 1.58; α was 1.83 (very large, according to Walter et al. (2009)); $\beta_{bulk}$ was -0.76º m$^{-1}$, suggesting backing; and $\beta_{total}$ was 0.76º m$^{-1}$. This case underscores challenges with any NTF. Because the nacelle speed was actually larger than the lidar speed for this case and the NTF was created under the mean case assumption that the lidar speed is greater than the nacelle speed, the addition of our NTF caused $\Delta$REWS$_{N-NTF}$ and $\Delta$REWS$_{\theta,N-NTF}$ to be lower than they should be.

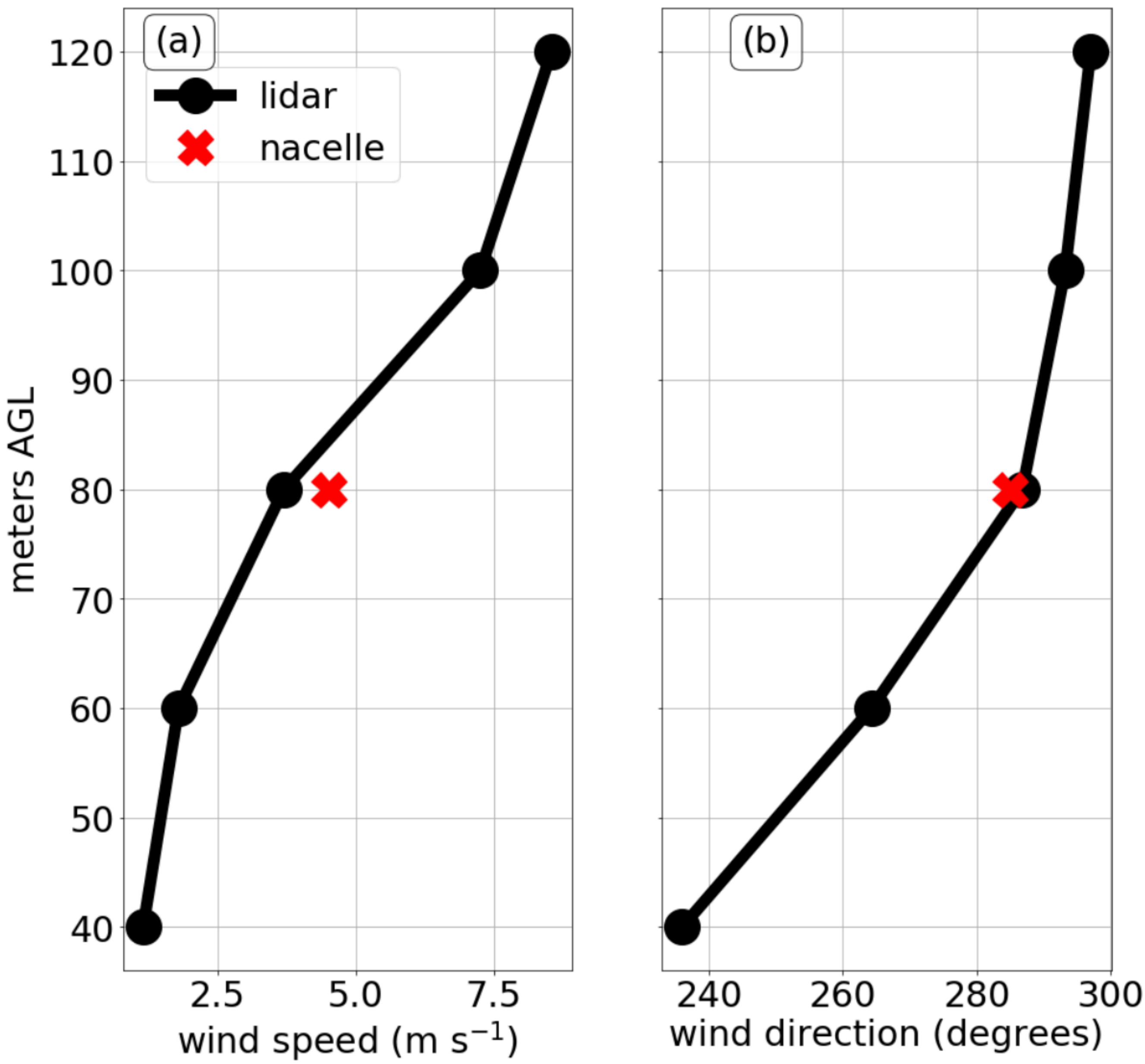

**Figure 7: Vertical profile of wind (a) speed, (b) direction during a case of strong shear. Black circle markers indicate the heights with lidar observations. The red X's denote the nacelle wind speed and direction during the case.**

Depending on which wind speed is used, the turbine power production for this case varies significantly, as calculated from Eq. 1 and the variable wind-speed dependent $C_p$ values of Schmitz (2015), interpolated to 0.01 m s$^{-1}$ bins. The air density is disregarded so as to not reveal the elevation of the test site. Instead, power is expressed as a percentage of rated. These powers

are meant only as example values as a simple power curve created from basic principles and does not surmise the real, more complicated, power curve.

The lidar wind speed suggests a power 4.7% of rated, the nacelle wind speed suggests a power 8.4% of rated, the REWS suggests a power 14% of rated, and the REWS$_\theta$ suggests a power 13.4% of rated (Table 2). For this case, the discrepancies of power are ~10% of rated power simply because of the different wind speed assessments. Although exact turbine power production cannot be given for this time, REWS$_\theta$ and REWS are the most accurate metrics to the actual power production, but still vary from it somewhat.

| *Wind shear metric* | *Equation* | *Eq. #* |
|---|---|---|
| $REWS$ | $\sqrt[3]{\frac{\sum_{i=0}^{3} A_{z(i)\ to\ z(i+1)} \left(\frac{U_{z(i+1)} + U_{z(i)}}{2}\right)^3}{A}}$ | (10) |
| $\Delta REWS_{N-NTF}$ | $\mathrm{REWS} - U_{nacelle} - NTF$ | (11) |
| $\Delta REWS_L$ | $\mathrm{REWS} - U_{lidar}$ | (12) |
| $REWS_\theta$ | $\sqrt[3]{\frac{\sum_{i=0}^{3} A_{z(i)\ to\ z(i+1)} \left(\frac{U_{z(i+1)}\cos(\Delta\theta_{z(i+1)}) + U_{z(i)}\cos(\Delta\theta_{z(i)})}{2}\right)^3}{A}}$ | (13) |
| $\Delta REWS_{\theta,N-NTF}$ | $\mathrm{REWS}_\theta - U_{nacelle} - NTF$ | (14) |
| $\Delta REWS_{\theta,L}$ | $\mathrm{REWS}_\theta - U_{lidar}$ | (15) |
| $\alpha$ | $\frac{\log\left(\frac{U_{top}}{U_{bottom}}\right)}{\log\left(\frac{z_{top}}{z_{bottom}}\right)}$ | (16) |
| $\beta_{bulk}$ | $\frac{\theta_{top} - \theta_{bottom}}{z_{top} - z_{bottom}}$ | (17) |
| $\beta_{total}$ | $\frac{\sum_{i=0}^{3} \lvert \theta_{z(i+1)} - \theta_{z(i)} \rvert}{z_{top} - z_{bottom}}$ | (18) |

**Table 1. Summary of shear metrics**

| *Wind speed metric* | *Wind speed (m s$^{-1}$)* | *Power (% of rated)* |
|---|---|---|
| *Lidar* | 3.7 (m s$^{-1}$) | 4.7% |
| *Nacelle* | 4.5 (m s$^{-1}$) | 8.4% |
| *REWS* | 5.36 (m s$^{-1}$) | 14.0% |
| $REWS_\theta$ | 5.28 (m s$^{-1}$) | 13.4% |

**Table 2: Theoretical percent of rated power from interpolated Schmitz power curve and observed wind speeds**

### 3.2 Power curve calculation

For each shear metric, we calculated three power curves by segregating the actual 1-Hz power production recorded by the turbine's SCADA system (rather than using an idealized curve) into 0.5-m $s^{-1}$ wind speed bins. The three power curves are designated as such: a mean power curve (all the power data in the bin), a high case power curve (all the powers such that the shear metric at the time index of the power is greater than a certain critical value of the shear metric) and a low case power curve (all the powers such that the shear metric at the time index of the power is less than a certain critical value of the shear metric). The critical values are determined in Section 4.1.

Around the shear metric-based power curves, 99% confidence intervals were calculated using a two-tailed t-test at each bin following the confidence interval given in Eq. 9. The mean power curve (regardless of shear conditions) is considered to be the overall population mean for power production, $\mu$, so a confidence interval is not placed around the data.

Two different independent variables (wind speeds) can apply to our data set, the lidar wind speed at 80 m (L) and the nacelle wind speed offset by the NTF (N-NTF). For the $\Delta \text{REWS}_L$ case, the lidar wind speed (L) is used as the x-axis. For the other plots, the N-NTF is used for the x-axis. If the wrong wind speeds are used for the ΔREWS case power curves, the case means tend to collapse onto the mean power curve.

Additionally, differences between the overall mean power curve and the shear metric-based power curves were plotted. The confidence intervals on these plots come from the subtraction of the mean power curve from the bounds of the confidence intervals.

## 4 Results

Sections 4.1–4.3 describe distributions of shear metrics, determinations of critical values of the metrics, and correlations between the metrics. Sections 4.4–4.10 describe how the shear metric cases affect power production.

### 4.1 Histogram distributions of shear metrics and determination of critical values

Histograms and cumulative distribution functions of the shear metrics suggest a range of stability and shear conditions during the test period (Fig. 8). In Fig. 8, the histograms and the cumulative distribution functions are normalized separately so that the maximum value of each respective plot is 1.

The differences between $\Delta \text{REWS}_{N-NTF}$ and $\Delta \text{REWS}_L$, Figures 8a and b, emphasize the difference between the lidar and nacelle measurements of hub-height wind speed as well as the role of integrating the winds across the rotor disk. Although

$\Delta\text{REWS}_{N-NTF}$ (Fig. 8a) exhibits a wide distribution, $\Delta\text{REWS}_{L}$ (Fig. 8b) is centered more tightly around zero, likely because the REWS is calculated from lidar values and some variation in the wind occurs between the lidar and the nacelle. The critical value used for $\Delta\text{REWS}_{N-NTF}$ is 0, which segregates data with REWS greater than the offset nacelle wind speed ($0 < \Delta\text{REWS}_{N-NTF}$) and those with REWS less than the offset nacelle wind speed ($\Delta\text{REWS}_{N-NTF} < 0$). Likewise, the critical value used for $\Delta\text{REWS}_{L}$ is 0. Low $\Delta\text{REWS}_{N-NTF}$ cases make up 51.6% of the data, while high cases make up 48.4%. For $\Delta\text{REWS}_{L}$, low cases make up 49.8% of the data and high cases make up 50.2%. Neither of the $\Delta\text{REWS}_{\theta}$ cases (N-NTF and L) appear because the respective N-NTF and L histograms are nearly identical to their ΔREWS counterparts.

The distribution of $\alpha$ (Figure 8c) shows that winds tend to increase with height but that some cases of winds decreasing from 40 m to 120 m do occur, similar to Walter et al. (2009). To segregate between high and low values of $\alpha$, we use a threshold for high of 0.2 (as in Vanderwende and Lundquist (2012) and Wharton and Lundquist (2012b)) and a low threshold of 0.1 (same as Vanderwende and Lundquist (2012) and slightly greater than Wharton and Lundquist (2012b) who use 0.09). High cases of $\alpha$ make up 37.4% of the data and low cases comprise 40.7% of the data.

Just as with $\Delta\text{REWS}_{N-NTF}$ and $\Delta\text{REWS}_{L}$, a nearly 50-50 split of the $\zeta$ segregation occurs (Fig. 8d). The critical value is chosen to be 0, to split stable and unstable cases from each other, as explained in Section 2.2. Stable cases make up 52.8% of the data while convective cases make up 47.2% of the data. Only $\zeta$ values between -100 and 100 are shown in Fig. 8d to resolve most of the data.

The $\beta_{bulk}$ distribution (Fig. 8e), divided between veering ($\beta_{bulk} > 0$) and backing ($\beta_{bulk}$), shows a surprising prevalence of backing conditions, in contrast to other observations (Walter et al., 2009; Bodini et al., 2019b; Sanchez Gomez and Lundquist, 2019). Veer occurs 34.7% of the time while backing occurs 64.9% of the time. We suspect that the complex nature of the local terrain and/or the prevalence of cold front passages during this summertime period supports more backing than veering.

The $\beta_{total}$ distribution (Fig. 8f) is effectively an absolute-valued $\beta_{bulk}$ with an increased amount of low values because of occurrences of nonmonotonic shear. For $\beta_{total}$, the choice of 0.15 as a critical value was chosen experimentally by splitting the histogram of $\beta_{total}$ by varying the parameter of the critical value. Using 0.15 splits the data almost in half. The low $\beta_{total}$ case accounts for more than 54.1% of the filtered data and the high $\beta_{total}$ case accounts for more than 45.6% of the filtered data. Values other than 0.15° m$^{-1}$ were explored, such as 0.1 and 0.2° m$^{-1}$. Similar results were found with 0.1° m$^{-1}$, but with wider confidence intervals on the high $\beta_{total}$ case that lead to less significance. The 0.2° m$^{-1}$ case was also similar to the 0.15° m$^{-1}$ case, with worse symmetric divisions between high and low.

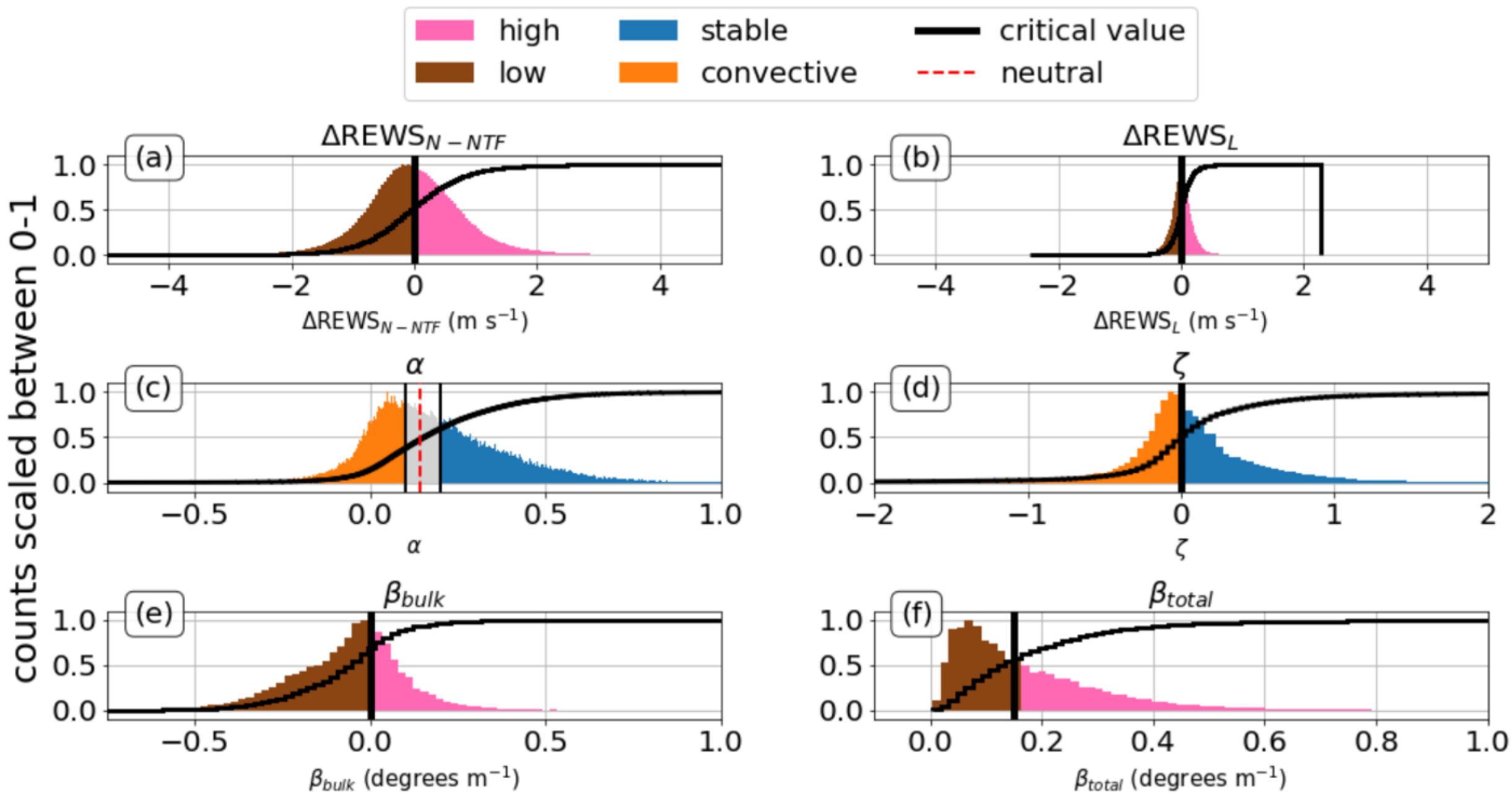

**Figure 8: Histograms and cumulative distribution functions (black curves) of metrics. Black vertical lines denote critical values and divide each shear metric into a high and low case. The number of bins used is different for most plots and the values were chosen experimentally. (a) ΔREWS$_{N-NTF}$ with a critical value of 0 with 400 bins. (b) ΔREWS$_L$ with a critical value of 0 with 400 bins. (c) $\alpha$ such that the low case is cut-off at 0.1, the high $\alpha$ case begins at 0.2, and the classic neutral value of $\alpha$ is shown at 1/7 (red dashed line) with 1000 bins. (d) $\zeta$ with a critical value of 0 with 5000 bins. (e) $\beta_{bulk}$ with a critical value of 0 with 200 bins. (f) $\beta_{total}$ with a critical value of 0.15 with 300 bins. Outlier values are not plotted to reduce visual clutter.**

### 4.2 Polar distributions of shear metrics

To explore variations of the metrics with wind direction, we created polar plots for each shear metric (Fig. 9) by binning data into 5º bins using the lidar wind direction and plotting the median of the data in the bins. Medians were chosen rather than means to account for the long tails on measurements, such as $\zeta$ and $\beta_{bulk}$.

For ΔREWS$_{N-NTF}$ and ΔREWS$_L$ cases (and those including direction, not shown), a strong variation with wind direction occurs (Fig. 9a,b). Nearly all northerly wind direction bins are low ΔREWS cases and nearly all southerly wind direction bins are high ΔREWS cases. This variation with wind direction seems to arise from the terrain, with extremely complex terrain to the south because of an escarpment, and relatively flat terrain to the north (compared to the escarpment).

Similarly, $\alpha$ varies strongly with wind direction (Fig. 9c), though the variation is not as distinct as that of the ΔREWS cases. All the southerly wind directions are stable except for the south to south-southeasterly neutral cases. Northerly flow is typically neutral, with one convective point on the data boundary to the west-northwest and a cluster of convective data ranging from northerly to the north-northeasterly. The north-northeasterly directions are the ones with the lowest terrain

elevation change of any direction, while the topography just upwind of the equipment to the west-northwest and east-northeast actually descends before the turbine.

Stability, as defined by surface-layer scaling parameter $\zeta$ (Fig. 9d), resembles that defined by $\alpha$ (Fig. 9c). All southerly cases are stable except one (on the boundary of south-southeasterly flow), and some northerly directions are stable as well. However, the majority of the data with northerly flow are convective. North-northeasterly winds are convective (as with convective $\alpha$) though some westerly convective points occur, which are not seen with $\alpha$. However, stable points still exist to the north, generally with westerly components. This distribution could be a result of the plateau's (mainly southerly) escarpment wraps around the turbines to the west somewhat. Because $\zeta$ involves friction velocity, this terrain could be enough to shear the flow and cause $\zeta$ to be stable to those directions. However, this might not be the case because the terrain is not enough to cause westerly REWS metrics to increase.

$\beta_{bulk}$ does not show a strong directional dependence: nearly all directions have median low $\beta_{bulk}$ values, which implies a uniform dominance of backing winds (Fig. 9e). However, the west to west-northwest values are high, and therefore generally positive, which implies a dominance of veering winds from those directions. Given how few winds come from west-northwest, proposing a mechanism for this veering is difficult.

The directional distribution of $\beta_{total}$ is somewhat similar to that of $\alpha$ and $\zeta$, where lower values of $\beta_{total}$ occur under directions of convection (as denoted by $\alpha$ and $\zeta$) and greater values of $\beta_{total}$ happen under directions of stability (Fig. 9f). However, not all stable directions correspond to high $\beta_{total}$ and not all convective directions correspond to low $\beta_{total}$. These results are somewhat expected and physically reasonable because the lack of convection during the night allows the atmosphere to decouple with height, increasing veering or backing. However, these results are not as directionally consistent as for $\beta_{bulk}$.

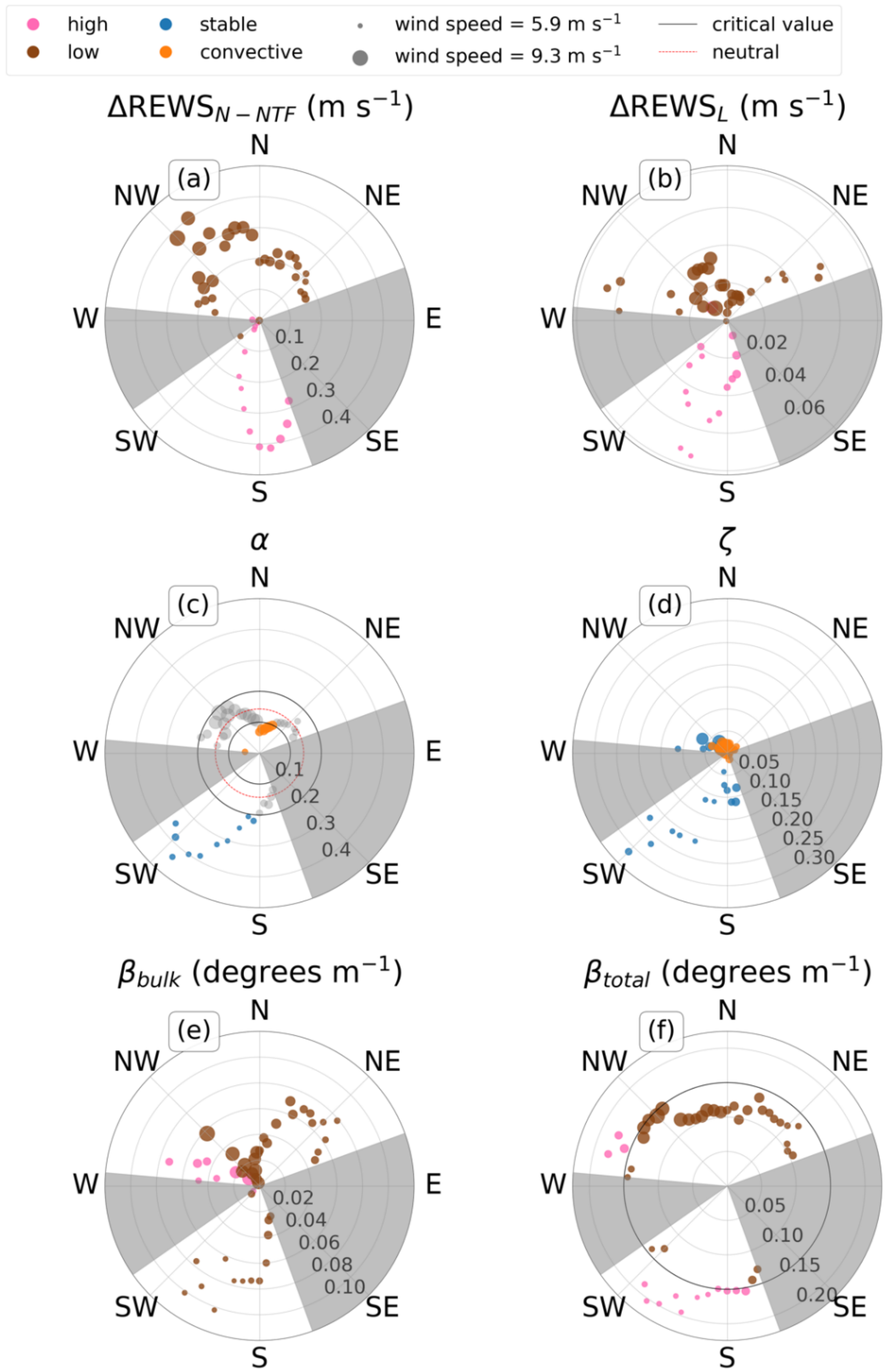

**Figure 9: Polar median distributions of metrics for 5º direction bins. Black circles denote nonzero critical values and divide each shear metric into a high and low case. In the case where the metric could take on negative values, the negative values were wrapped to positive but colored following Fig. 8. (a) ΔREWS$_{N-NTF}$ with a critical value of 0. (b) ΔREWS$_L$ with a critical value of 0. (c) such that convective cases have $\alpha < 0.1$, stable cases have $\alpha > 0.2$, and the industry-standard neutral value of $\alpha$ is shown at 1/7 (red dashed**

**line) and moderate $\alpha$s are gray. (d) $\zeta$ with a critical value of 0. (e) $\beta_{bulk}$ with a critical value of 0. (f) $\beta_{total}$ with a critical value of 0.15.**

### 4.3 Temporal distributions of shear metrics

To find variations of the metrics with time, each shear metric is binned by local time hour and the median of the data in each hour bin is plotted (Fig. 10). Medians were again chosen rather than means to account for the long tails on certain measurements such as $\zeta$ and $\beta_{total}$.

Temporally, neither $\Delta\text{REWS}_{N-NTF}$ nor $\Delta\text{REWS}_{L}$ exhibit a clear diurnal cycle. Both high and low ΔREWS periods occur during both daytime and nighttime hours (Fig. 10a,b). Additionally, the two cases do not covary with each other by hour, as the L case changes sign between high and low eight times while the NTF-shifted nacelle wind speed case only changes sign four times. The times at which the sign changes between the two cases are not always the same. However, when the two cases do change signs at the same times (04:00–05:00,15:00–16:00), the sign changes at those times are always the same.

A clear diurnal cycle manifests for $\alpha$ (Fig. 10c), with stable values at night decreasing to neutral values during the morning transition and convective values during the day. During the evening transition, neutral values reoccur with stable cases reemerging later at night. The morning transition takes longer than the evening transition because solar heating requires a few hours to heat the ground enough to begin convection (Lapworth, 2005; Lapworth, 2009). Once the sun begins to set, most of the remaining heat from the ground is lost quickly because of convection, leaving the ground to cool radiatively (on a clear night), meaning the evening transition should be relatively rapid (Lee and Lundquist, 2017). Like $\alpha$, $\zeta$ shows a strong temporal cycle (Fig. 10d). During daytime hours $\zeta$ becomes negative (convective) and during nighttime hours $\zeta$ becomes positive (stable).

Previous investigations of stability metrics for wind energy studies have relied on $\alpha$ as a stability metric (Wharton and Lundquist, 2012b; Vanderwende and Lundquist, 2012). We break up our $\alpha$ data based on those stability delineations and see that $\alpha$ does have a strong daily cycle, which would be expected for a stability metric in such a location, and refer to high and low $\alpha$ cases as stable and convective, respectively, to match with prior research. However, directionally, there appears to be a strong influence of terrain on stability. Thus, untangling the interaction between complex terrain and stability is challenging in this location.

$\zeta$ is treated in a similar manner as $\alpha$. A strong diurnal cycle emerges, which is to be expected; however, the directional variation is dominated by stable cases from directions that could likely be influenced by topography. Because the Obukhov Length calculation incorporates friction velocity, it  (and thus $\zeta$) is clearly influenced by the terrain at this location.

The diurnal cycle also emerges in $\beta_{bulk}$ (Fig. 10e). All hours have median low $\beta_{bulk}$ values which implies a dominance of backing winds at all times of the day at this complex terrain site. No hours exhibit a median veer. However, the backing is weaker (less negative) during the convective hours (as also suggested by $\alpha$ and $\zeta$). This behavior is physically reasonable because convective eddies mix momentum through the boundary layer, coupling winds throughout the boundary layer, such that the wind direction should vary little with height during convection.

As explained earlier, $\beta_{total}$ is effectively the absolute value of $\beta_{bulk}$ (but with a nonzero critical value of 0.15), and so the temporal distribution of $\beta_{total}$ (Fig. 10f) somewhat resembles that of $\beta_{bulk}$ (Fig. 10e). Stronger veer dominates from midnight until 08:00 local time, likely because of nocturnal decoupling. The overall temporal distribution of veer appears in sync with the temporal distribution of $\alpha$; however, the choice of the critical value of 0.15 (the choice of which is explained in Section 4.1) affects the visualization of this distribution.

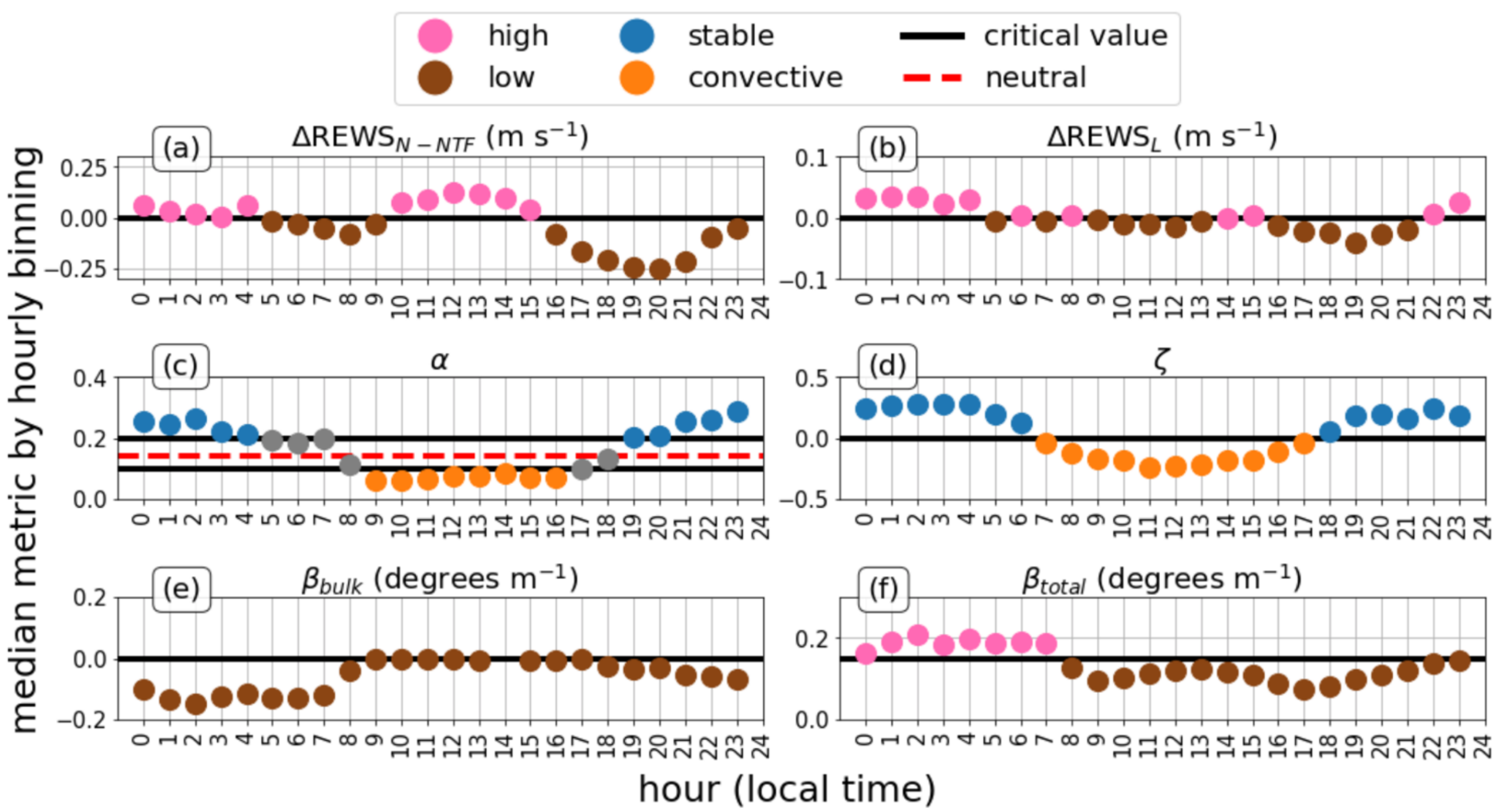

**Figure 10: Temporal median distributions of metrics for hourly bins. Black lines denote critical values and divide each shear metric into a high and low case. (a) ΔREWS$_{N-NTF}$ with a critical value of 0. (b) ΔREWS$_L$ with a critical value of 0. (c) $\alpha$ such that the convective cases are $\alpha < 0.1$, stable cases are $\alpha > 0.2$, and the industry-standard neutral value of $\alpha$ is shown at 1/7 (red dashed line) and "neutral" $0.1 < \alpha < 0.2$ are grey. (d) $\zeta$ with a critical value of 0. (e) $\beta_{bulk}$ with a critical value of 0. (f) $\beta_{total}$ with a critical value of 0.15.**

### 4.4 Further comparisons between selected metrics

While the median diurnal cycle suggests a relationship between $\boldsymbol{\zeta}$ and $\boldsymbol{\alpha}$, we would like more robust evidence of this correlation. To find such a correlation, we computed linear correlation coefficients between $\boldsymbol{\zeta}$ and $\boldsymbol{\alpha}$ across 5º lidar direction bins treating each 5° wind direction bin separately because of the influence of terrain on the location. However, after calculating correlations of metrics within these 5° wind direction bins, we found little evidence of agreement between these metrics. The strongest linear correlation values between $\boldsymbol{\zeta}$ and $\boldsymbol{\alpha}$ are only 0.4, these values occur in the southerly bins. The maximum linear correlation between $\boldsymbol{\zeta}$ and $\boldsymbol{\alpha}$ for northerly bins is less than 0.2, indicating very poor correlation. We applied the same directional binning linear correlation method to both types of **ΔREWS** and $\boldsymbol{\zeta}$ and both types of **ΔREWS** and $\boldsymbol{\alpha}$. No combinations had greater correlations than 0.18 for any direction bin (figures not shown). This lack of any directional correlation further suggests that the metrics do not map directly to atmospheric stability metrics in this region with complex terrain.

Additionally, because the histograms of the directional and nondirectional REWS metrics are so similar (Section 4.1), nondirectional and directional ΔREWS power curves strongly resemble each other. Power curves based on $\Delta\text{REWS}_{\theta,N-NTF}$ and $\Delta\text{REWS}_{\theta,L}$ are not statistically significantly different from that of $\Delta\text{REWS}_{N-NTF}$ and $\Delta\text{REWS}_{L}$ respectively, so only results for $\Delta\text{REWS}_{N-NTF}$ and $\Delta\text{REWS}_{L}$ are shown (Section 4.5 and Section 4.6, respectively). The greatest differences between the directional and nondirectional REWS metrics occur at high yaw misalignments, suggesting that a general yaw misalignment is more impactful than any further veer across the rotor disk under the specific conditions our location faced.

### 4.5 $\Delta\textbf{REWS}_{N-NTF}$ impacts on power production

$\Delta\text{REWS}_{N-NTF}$ shows statistically significant differences in actual turbine power production during cases of high $\Delta\text{REWS}_{N-NTF}$ (generally high shear) and low $\Delta\text{REWS}_{N-NTF}$ (generally low shear or negative shear) (Fig. 11). The difference between the metrics is greatest around 7.5 m s$^{-1}$ and 12.5 m s$^{-1}$, as measured by the NTF-shifted nacelle wind speed.

Further, power production during high $\Delta\text{REWS}_{N-NTF}$ conditions significantly exceeds the mean power production for conditions with NTF-shifted nacelle wind speeds between 3.19 m s$^{-1}$ and 13.70 m s$^{-1}$ (Fig. 11). Generally, increases range from around 20 kW to 40 kW but can exceed 60 kW (2.7% to 4% of rated) (Fig. 11b). The maximum average increase of power from the mean in the significant range is between 45.73 kW and 60.44 kW (3% to 4% of rated) at 12.70 m s$^{-1}$.

Power production during low $\Delta\text{REWS}_{N-NTF}$ conditions is significantly less than the mean power production for NTF-shifted nacelle wind speeds between 2.20 m s$^{-1}$ and 13.70 m s$^{-1}$ (Fig. 11b). The maximum average decrease of power from the mean in that range is between 28.20 kW and 29.27 kW (1.9% to 2% of rated), which occurs at the NTF-shifted nacelle wind speed of 7.70 m s$^{-1}$ (Fig. 11b).

Although the impact on power is somewhat symmetric, the high $\Delta\text{REWS}_{N-NTF}$ case leads to greater increases than the decreases of the low $\Delta\text{REWS}_{N-NTF}$ case at high NTF-shifted nacelle wind speeds above 8 m s$^{-1}$ or so.

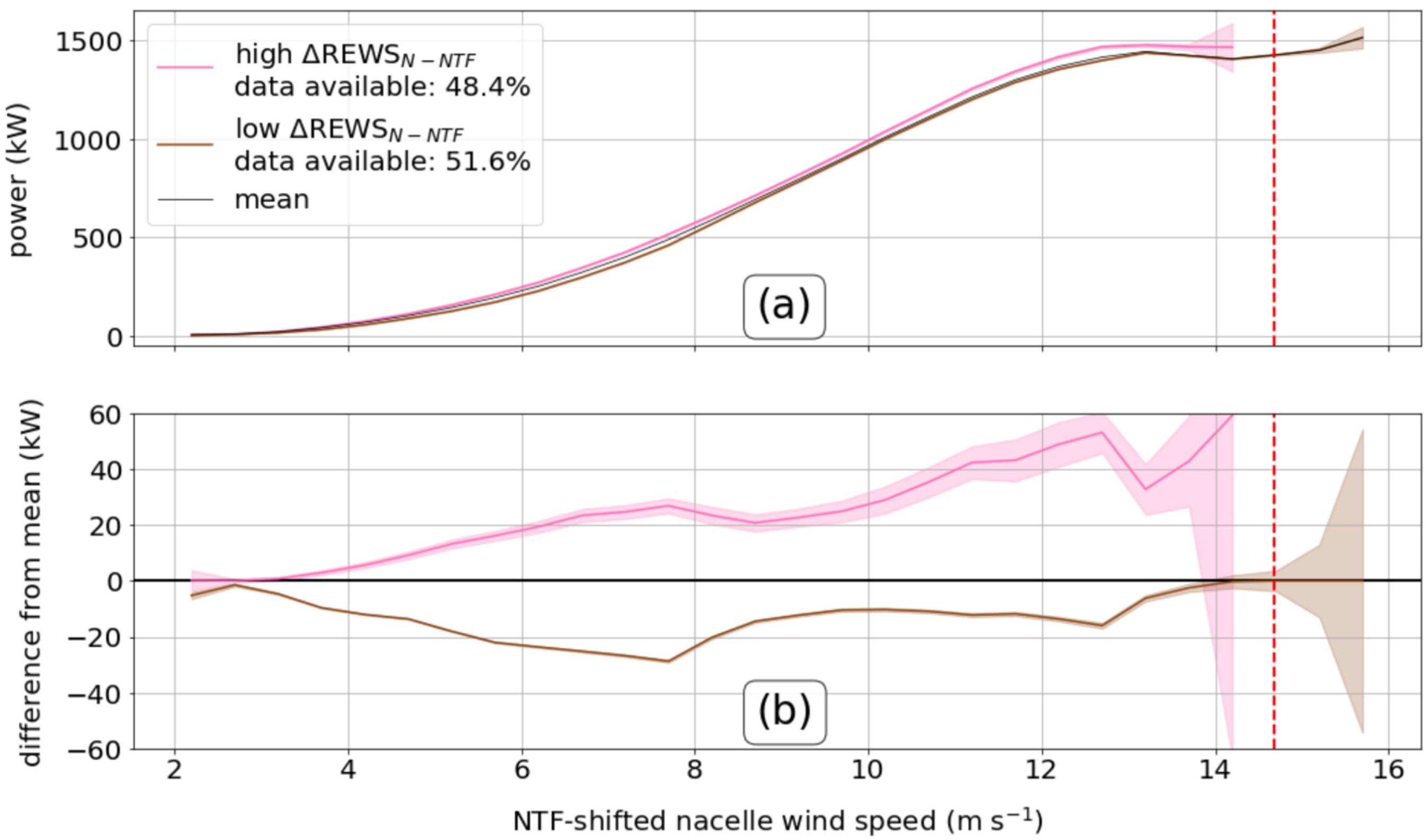

**Figure 11: (a) 99% confidence interval power curves generated for both $\Delta\text{REWS}_{N-NTF}$ cases. The mean power curve is shown by the solid black line. (b) Difference between two $\Delta\text{REWS}_{N-NTF}$ cases and the mean power curve where an overlap with 0 shows insignificance. The red dashed line corresponds to the nacelle-rated wind speed of 14 m s$^{-1}$ but is shifted up because of the NTF-shifted nacelle wind speed being offset from the nacelle wind speed. The high uncertainty above rated nacelle wind speeds is an artifact of low data availability and curtailment at rated speeds that we were unable to filter out.**

### 4.6 $\Delta\text{REWS}_L$ impacts on power production

Actual turbine power production during high and low $\Delta\text{REWS}_L$ conditions varies significantly, showing statistically significant differences between high and low cases (Fig. 12). The difference between the metrics is greatest around 11 m s$^{-1}$. Further, power production during high-$\Delta\text{REWS}_L$ (typically high-shear) conditions is significantly greater than the mean power production for conditions with hub-height lidar wind speeds between 4.07 m s$^{-1}$ and 12.57 m s$^{-1}$ (Fig. 12b). However, just as with $\Delta\text{REWS}_L$, that difference varies depending on the hub-height lidar wind speed. Increases in power, compared to the mean power curve, generally range from around 5 kW to 40 kW (0.3% to 2.7% of rated) but can exceed 70 kW (4.7% of

rated) (Fig. 12b). The maximum average increase of power from the mean in the significant range is between 31.08 kW and 74.86 kW (2.1% to 5% of rated) at 11.07 m s$^{-1}$.

In contrast, power production during low-$\Delta REWS_L$ (typically low-shear or negative-shear) conditions is significantly less than the mean power production with hub-height lidar wind speeds between 3.07 m s$^{-1}$ and 12.57 m s$^{-1}$ (Fig. 12). The maximum average decrease of power from the mean in that range is between 22.56 kW and 25.10 kW (1.5% to 1.7% of rated), which occurs at 9.57 m s$^{-1}$ (Fig. 12b). At high lidar wind speeds above 8 m s$^{-1}$ or so, the high-$\Delta REWS_L$ case leads to greater increases than the decreases of the low-$\Delta REWS_L$ case.

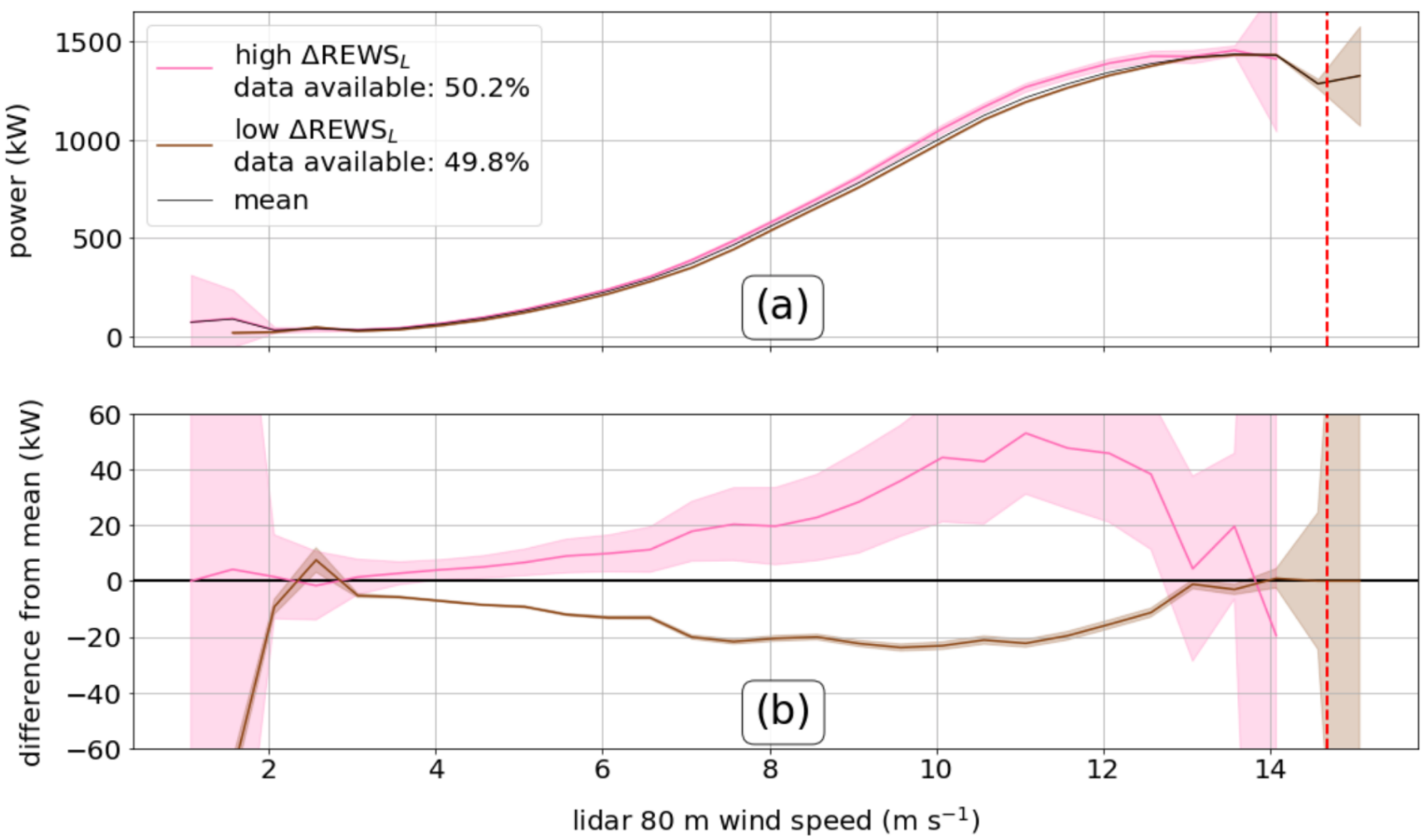

**Figure 12: (a) 99% confidence interval power curves generated for both $\Delta REWS_L$ cases. The mean power curve is shown by the solid black line. (b) Difference between two $\Delta REWS_L$ cases and the mean power curve where an overlap with 0 shows insignificance. The red dashed line corresponds to the nacelle rated wind speed of 14 m s$^{-1}$ but is shifted up because of the NTF-shifted nacelle wind speed being offset from the lidar wind speed. The high uncertainty above rated nacelle wind speeds is an artifact of low data availability and curtailment at rated speeds that we were unable to filter out.**

### 4.7 $\alpha$ impacts on power production

Turbine power production does not vary clearly as a function of $\alpha$ (Fig. 13), suggesting that $\alpha$ is not a powerful metric for assessing power production at this site. The low $\alpha$ case shows significantly greater power production than the high $\alpha$ case for nearly all NTF-shifted nacelle wind speeds between around 8 m s$^{-1}$ to 12.5 m s$^{-1}$. The high $\alpha$ case generates significantly less

power than the mean by 5 kW to 20 kW (0.3% to 1.3% of rated) for wind speeds from around 8 m s$^{-1}$ to 12.5 m s$^{-1}$ (Fig. 13). The maximum average decrease of power from the mean in that range is between 10.15 kW and 19.15 kW (0.7% to 1.3% of rated), which occurs at the NTF-shifted nacelle wind speed of 11.20 m s$^{-1}$ (Fig. 13b). The low $\alpha$ case generates significantly greater power than the mean by around 1 kW to 20 kW (0.1% to 1.3% of rated) from around 8 m s$^{-1}$ to 13 m s$^{-1}$. The maximum average increase of power from the mean in that range is between 17.29 kW and 20.58 kW (1.2% to 1.4% of rated), which occurs at the NTF-shifted nacelle wind speed of 12.20 m s$^{-1}$ (Fig. 13b). However, at lesser wind speeds (below 8 m s$^{-1}$), both the high and low $\alpha$ cases demonstrate inconsistent oscillatory variability and even switch sign with each other at NTF-shifted nacelle wind speeds just past the cut-in wind speed. Some significant wind speed cases exist below 8 m s$^{-1}$; however, the differences in power from the mean are very small.

This inconsistent and unsatisfying picture of the utility of $\alpha$ in predicting power production led us to experiment with changing the threshold critical $\alpha$ values. Setting a smaller low bound (reducing the number of convective cases) only increases significance in Fig. 13b until an $\alpha$ of 0.07, but that $\alpha$ threshold fails to match the diurnal cycle. As such, the original critical low bound of 0.10 is used. Setting a lower low threshold than 0.10 or a higher high threshold than 0.20 does not enhance differences between the metrics and the means. Rather, the confidence intervals widen, because of fewer low or high data points, while the mean values do not change, leading to insignificance. Furthermore, because of the preponderance of neutral $\alpha$ values, only 78.2% of the filtered data set is used to create the high and low $\alpha$ curves. Neutral values are included in the mean power curve. However, changing our critical values (and thus placing neutral data into the high and low cases) leads to greater insignificance. The data for such insignificant results are not shown.

These results of $\alpha$ impacts on power production are somewhat counterintuitive to physically based expectations but are similar to the results of Vanderwende and Lundquist (2012), based on 2 months of data at this site several years previously. High $\alpha$ is a measurement of high shear, and high shear implies that the top of the turbine rotor disk is associated with a greater wind speed than the hub height, which should be associated with a greater wind speed than the bottom of the turbine rotor disk. However, the greater wind speeds near the top of the rotor disk may not be able to compensate enough for the lesser wind speeds near the bottom of the rotor disk because of complicated wind profiles that result from the locally complex terrain. The greater wind speeds near the top of the rotor disk also may not be orthogonal to the rotor disk, because of veering or backing, and therefore cannot be harvested efficiently by the turbine blades.

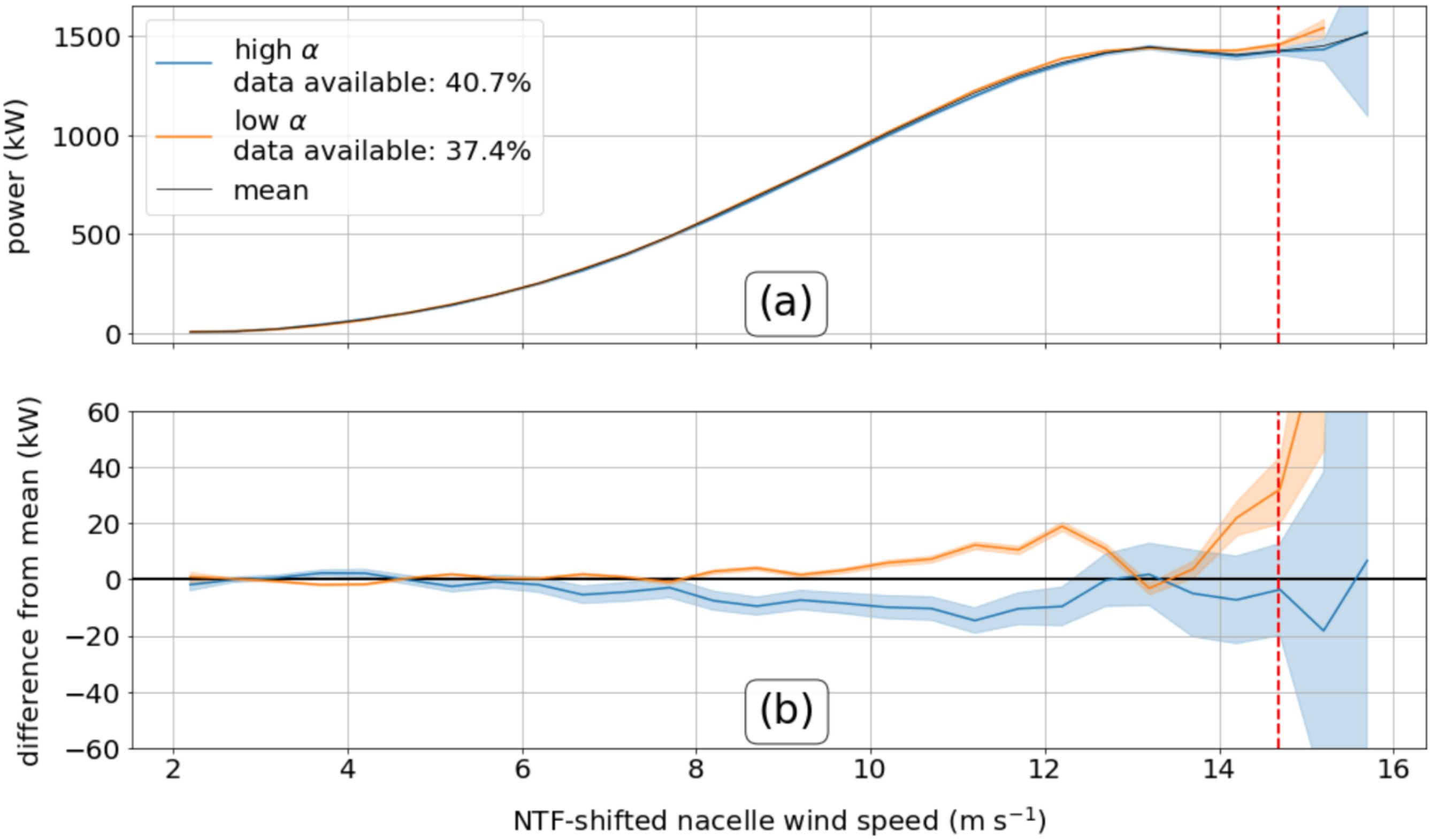

**Figure 13: (a) 99% confidence interval power curves generated for both $\alpha$ cases. The mean power curve is shown by the solid black line. (b) Difference between two $\alpha$ cases and the mean power curve where an overlap with 0 shows insignificance. The red dashed line corresponds to the nacelle rated wind speed of 14 m s$^{-1}$ but is shifted up because of the NTF-shifted nacelle wind speed being offset from the nacelle wind speed. The high uncertainty above rated nacelle wind speeds is an artifact of low data availability and curtailment at rated speeds that we were unable to filter out.**

### 4.8 $\zeta$ impacts on power production

The impact of stability as quantified by $\zeta$ (Fig. 14) is more easily interpretable than that of $\alpha$ (Fig. 13), but is not as clear as that of the REWS metrics (Figs. 11,12), suggesting that $\zeta$ has some skill in assessing power production even though $\zeta$ is based on data collected near the surface.

The low $\zeta$ case, associated with daytime conditions, shows significantly greater power production than the high $\zeta$ case, associated with nighttime conditions, for nearly all NTF-shifted nacelle wind speeds between 4 m s$^{-1}$ and 13 m s$^{-1}$. The high $\zeta$ case generates significantly less power than the mean by around 1 kW to 20 kW (0.1% to 1.3% of rated) for wind speeds from around 4 m s$^{-1}$ to 12.5 m s$^{-1}$ (Fig. 14). The maximum average decrease of power from the mean in that range is between 2.49 kW and 18.18 kW (0.2% to 1.2% of rated), which occurs at the NTF-shifted nacelle wind speed of 12.20 m s$^{-1}$ (Fig. 14b). The low $\alpha$ case generates significantly greater power than the mean as well as significantly greater power than the high case by 1 kW to 16 kW (0.1% to 1.1% of rated) from 8 m s$^{-1}$ to 13 m s$^{-1}$. The maximum average increase of power from the

mean in that range is between 13.26 kW and 15.82 kW (0.9% to 1.1% of rated), which occurs at the NTF-shifted nacelle wind speed of 12.20 m s$^{-1}$ (Fig. 14b).

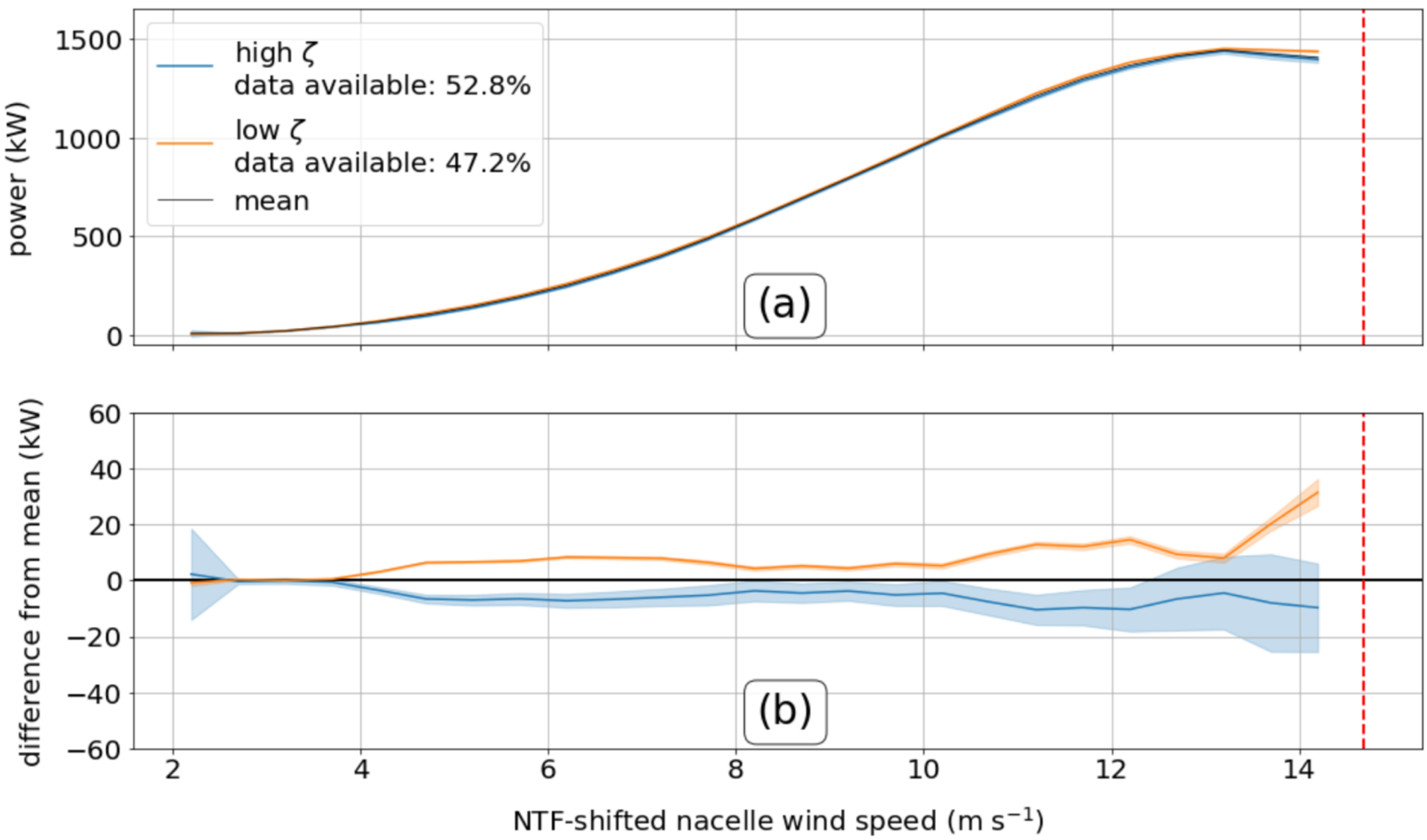

**Figure 14: (a) 99% confidence interval power curves generated for both $\zeta$ cases. The mean power curve is shown by the solid black line. (b) Difference between two $\zeta$ cases and the mean power curve where an overlap with 0 shows insignificance. The red dashed line corresponds to the nacelle rated wind speed of 14 m s$^{-1}$ but is shifted up because of the NTF-shifted nacelle wind speed being offset from the nacelle wind speed.**

### 4.9 $\beta_{bulk}$ impacts on power production

The influence of $\beta_{bulk}$ on turbine power production depends very closely on wind speed. Below 10 m s$^{-1}$, $\beta_{bulk}$ has almost wholly insignificant results $\alpha$ (Fig. 15). However, above that speed, small but significant oscillatory gains and losses in power occur. High $\beta_{bulk}$ (veering) leads to power gains, while low $\beta_{bulk}$ (backing) leads to power deficits. At wind speeds below rated, confidence bounds on the high $\beta_{bulk}$ case do not exceed 20 kW (1.3% of rated) of power increase, and confidence bounds on the low $\beta_{bulk}$ case do not exceed 10 kW (0.7% of rated).

The difference in power production seen between veer and backing at wind speeds above 10 m s$^{-1}$ resemble the results of Wagner et al. (2010). However, turbine yaw misalignment is not explicitly controlled for in our paper and only mean veer

and backing are examined, when different values could have different effects on power production. Additionally, values of $\beta_{bulk}$ tend to approach 0 for both high and low cases (Fig. 16). As such, the significant portions of the power curve above 10 m s$^{-1}$ are not a result of higher or lower values of $\beta_{bulk}$ occurring, but rather lower values of $\beta_{bulk}$ occurring with faster wind speeds. That greater wind speeds see less shear and veer is also physically reasonable because greater wind speeds tend to mechanically mix momentum through winds at all heights.

Finally, overall, nearly twice as much low $\beta_{bulk}$ data (veering) exists than high $\beta_{bulk}$ data (backing), remarkably different from other field campaigns in flat terrain (Walter et al., 2009; Sanchez Gomez and Lundquist, 2019) or offshore (Bodini et al., 2019b). This disparity suggests that the confidence intervals around the high (veer) case would be tightened with more data.

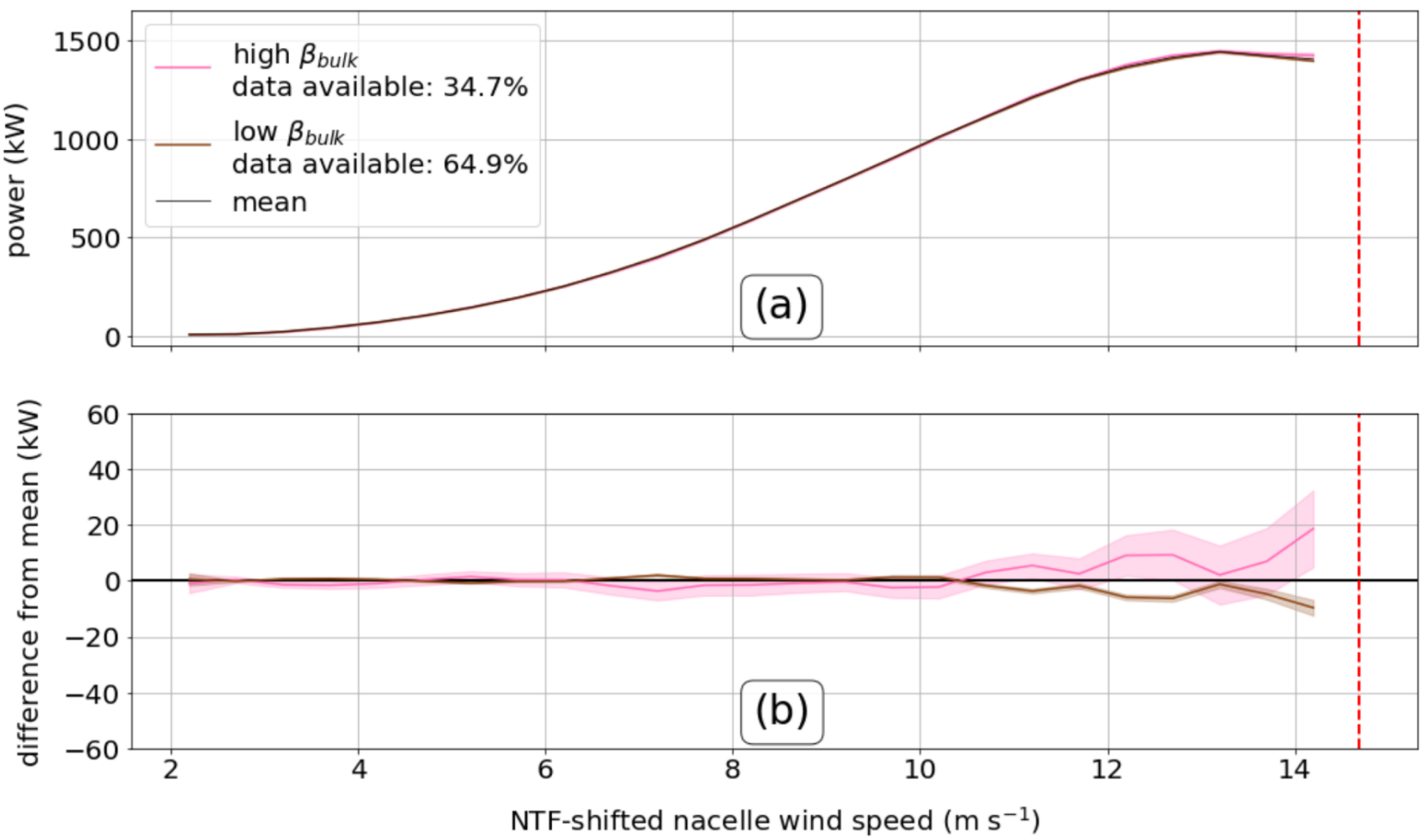

**Figure 15: (a) 99% confidence interval power curves generated for high and low $\beta_{bulk}$. The mean power curve is shown by the solid black line. (b) Difference between high and low $\beta_{bulk}$ and the mean power curve where an overlap with 0 shows insignificance. The red dashed line corresponds to the nacelle rated wind speed of 14 m s$^{-1}$ but is shifted up because of the NTF-shifted nacelle wind speed being offset from the nacelle wind speed.**

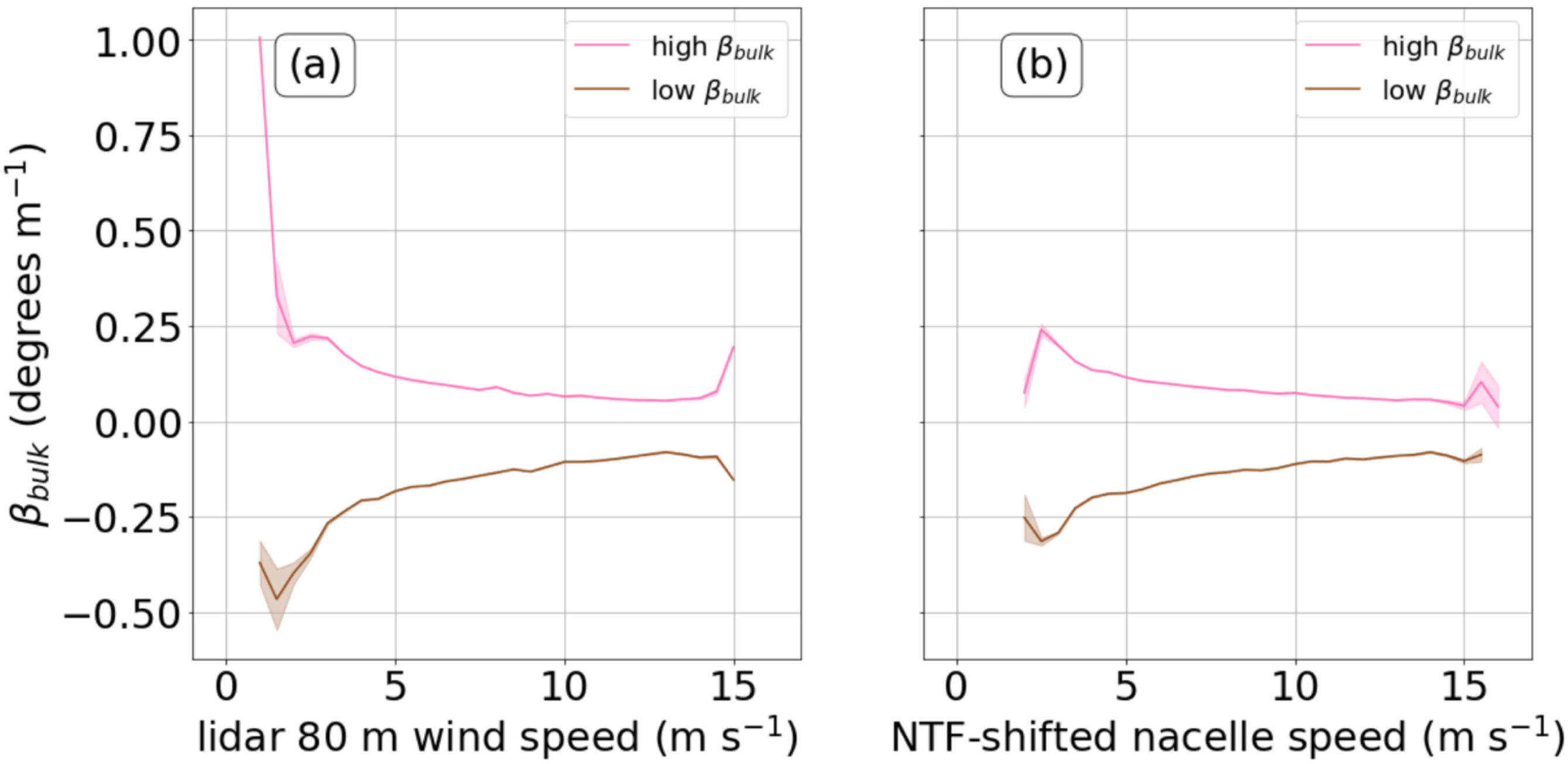

**Figure 16: 99% confidence interval mean high and low $\beta_{bulk}$ as a function of wind speed from (a) the lidar at 80 m and (b) the hub-height NTF-shifted nacelle wind speed. Means are used rather than medians to agree with means used for power curve plots and to put confidence intervals around the data.**

### 4.10 $\beta_{total}$ impacts on power production

Power gains and losses for $\beta_{total}$ exhibit differences between high directional veering/backing and low directional veering/backing from 4.5 m s$^{-1}$ to 12.5 m s$^{-1}$ (Fig. 17). Veering or backing undermines power production. Low values of $\beta_{total}$ imply a lack of directional shear across the turbine rotor disk, meaning that the winds across the rotor point orthogonally at the rotor plane and thus will not decrease power. Veering or backing reduces the magnitude of the winds orthogonal to the rotor disk, undermining power production. Low $\beta_{bulk}$ happens more often than high $\beta_{bulk}$ by a factor of nearly two. This lack of symmetry leads to a decrease in power production because low $\beta_{bulk}$ leads to a decrease in power production and high $\beta_{bulk}$ does not occur frequently enough to make up for it.

Additionally, high values of directional shear exert a greater impact on power production (just over 10 kW or 0.7% of rated) than low values of directional shear (which never exceeds 10 kW or 0.7% of rated). At greater wind speeds, the high $\beta_{total}$ case appears to lose even more power. This disparity is physically reasonable because the more the direction veers, the less power the turbine can extract from the atmosphere compared to a nonveered flow.

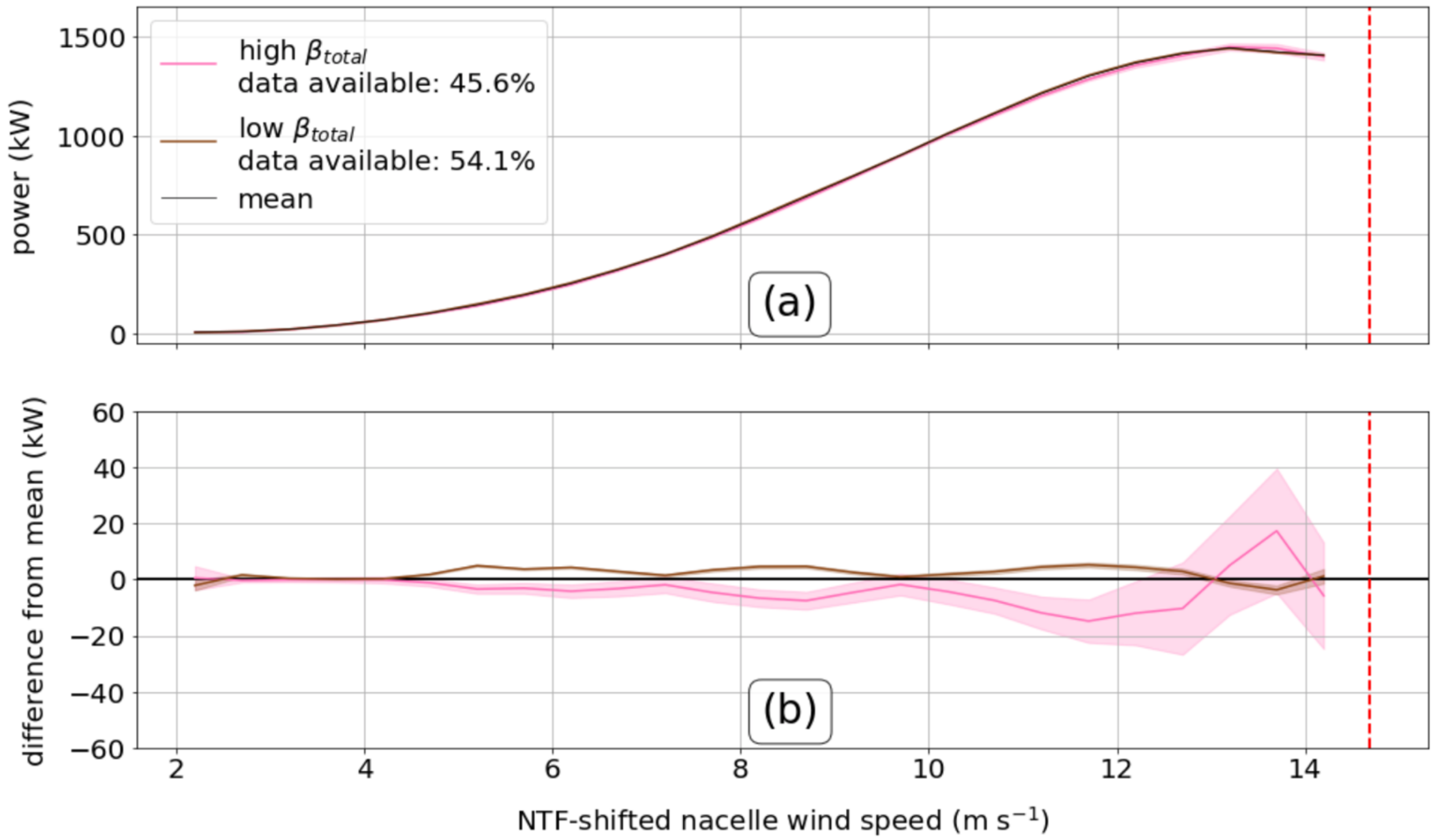

**Figure 17 (a) 99% confidence interval power curves generated for high and low $\beta_{total}$ cases. The mean power curve is shown by the solid black line. (b) Difference between two $\beta_{total}$ cases and the mean power curve where an overlap with 0 shows insignificance. The red dashed line corresponds to the nacelle rated wind speed of 14 m s$^{-1}$ but is shifted up because of the NTF-shifted nacelle wind speed being offset from the nacelle wind speed.**

## 5 Discussion and conclusions

In this article, we explore how wind shear, wind veer, and atmospheric stability impact actual power production of an operational megawatt-scale wind turbine at a commercial wind farm in the high plains of North America. SCADA systems measured the turbine's power productions at 1 Hz over a period of nearly 6 months. Additional measurements from a vertically profiling Doppler lidar and a meteorological mast allow us to derive wind shear and stability metrics $\Delta\text{REWS}_L$, $\Delta\text{REWS}_{N-NTF}$, $\alpha$, $\zeta$, $\beta_{bulk}$, and $\beta_{total}$.

After intercomparing these stability metrics, we use them to evaluate the power production in different regimes of shear by creating power curves for the different shear regimes. We evaluate power curves in terms of absolute changes in the power production of the turbine for the given regimes of shear. Percent changes (of rated power) are recorded as well.

REWS and its difference from hub-height wind speed from either the upstream lidar ($\Delta\text{REWS}_L$) or the nacelle anemometer ($\Delta\text{REWS}_{N-NTF}$) (Figs. 11,12) demonstrate the clearest impact of the wind profile on power production. These REWS-based metrics also rely on the most clear-cut bounds that could straightforwardly be applied to other turbines and wind farms. Significant differences between power curves created with REWS with and without direction (between $\text{REWS}_{\theta,L}$ and $\text{REWS}_L$ and between $\text{REWS}_{\theta,N-NTF}$ and $\text{REWS}_{N-NTF}$, respectively) do not occur at this site. However, small differences between REWS with and without direction do exist.

Both high ΔREWS cases ($0 < \Delta\text{REWS}_L$ and $0 < \Delta\text{REWS}_{N-NTF}$) lead to significantly greater power production than the mean power production (by up to 74.86 kW and 60.44 kW or 5% and 4% of rated, respectively) from lidar speeds of 4.07 m s$^{-1}$ to 12.57 m s$^{-1}$ and NTF-shifted nacelle wind speeds of 3.19 m s$^{-1}$ to 13.70 m s$^{-1}$, respectively. Both low ΔREWS cases ($\Delta\text{REWS}_L < 0$ and $\Delta\text{REWS}_{N-NTF} < 0$) lead to significantly less power production than the mean power production (by up to 25.10 kW and 29.27 kW or 1.7% and 2% of rated, respectively) from lidar speeds of 4.07 m s$^{-1}$ to 12.57 m s$^{-1}$ and NTF-shifted nacelle wind speeds of 3.19 m s$^{-1}$ to 13.70 m s$^{-1}$, respectively. The wind speed ranges where REWS is effective are the widest wind speed ranges of any of the metrics.

Although REWS is the most illuminating metric at this site, neither high nor low lidar or nacelle-based ΔREWS cases occur with a consistent temporal pattern through the data set (Figs. 10a,b). Terrain influences may dominate REWS at this site. High ΔREWS, quantified from both lidar-based and nacelle-based REWS, occurs more often during southerly flow (Figs. 9a,b), with inflow coming from low elevations up and over an escarpment, than for northerly flow, generally descending from higher terrain. Although this terrain influence is site specific, the REWS approach is likely more general and can be applied to other sites.

These results confirm the Sark et al. (2019) conclusion that measurement of REWS for power production purposes is necessary for complex terrain sites. Cost-benefit analyses are advised on the cost of implementation of installation and upkeep of inflow sensing equipment (like a Doppler lidar) to provide REWS measurements and the benefit of REWS for power production prediction. Of course, such equipment may be necessary for other purposes, such as adaptive alignment of turbines for wake control (Fleming et al., 2019).

Although previous results for power law coefficient $\alpha$ on power production (Wharton and Lundquist, 2012b; Vanderwende and Lundquist, 2012) suggest useful relationships, we find that, at this site, $\alpha$ results are too sensitive to chosen critical values and are not as clearly interpretable as the REWS results. For low $\alpha$ cases, significantly more power is produced than the mean around the middle of region II (from 8 m s$^{-1}$ to 12.5 m s$^{-1}$ or so) (Fig. 13). High $\alpha$ cases at nacelle speeds in that same portion of region II lead to significantly less power production than the mean (Fig. 13). However, at slower wind

speeds (below 8 m $s^{-1}$), these same results only apply to a lesser change in power production, and the two cases are often not significantly different from each other or the mean case. Part of the explanation of the muddled results is that $\alpha$ is only a measure of the shear, not of the actual wind speeds that comprise the inflow profile. Although the power curves are plotted as a function of the nacelle wind speed, this value may differ from the true wind speed at nacelle height and that speed may vary more over the rest of the rotor disk as well.

The power law coefficient $\alpha$ exhibits other weaknesses. Interestingly, wind speed shear $\alpha$ and wind direction veer in the form of $\beta_{bulk}$ and $\beta_{total}$ fail to show a clear relationship with each other at this location. Likewise, $\alpha$ does not correlate with REWS metrics or $\zeta$. Finally, $\alpha$ has the issue of data loss. Neutral conditioned data are not considered, meaning that around 22% of the filtered data was not used. In contrast, because of the clear demarcations for the REWS metrics, 100% of the REWS data could be used.

Additionally, these $\alpha$ results contrast somewhat with previous findings by Wharton and Lundquist (2012b). In a different site with channeled flow that could not exhibit veer, they found that high $\alpha$ increased power during wind speeds from 8 m $s^{-1}$ to 10 m $s^{-1}$. Although $\alpha$ does exhibit a strong daily cycle (convective in local daytime hours and stable at night), it also varies strongly with direction (stable when coming over very complex terrain, neutral otherwise, and convective when the fetch covers the flattest terrain). As such, the $\alpha$ in our case functions greatly as a descriptive indicator of inflow characteristics. This disparity in topography could account for the difference in findings.

However, our results agree well with those found by Vanderwende and Lundquist (2012), whose study used many more turbines over a shorter time period several years ago at this site. They assessed power curves with $\alpha$ bounds as well. They found that low $\alpha$ increases power at wind speeds in the higher wind speed portion of region II of the power curve, which generally follow our results between 8 m $s^{-1}$ and 12.5 m $s^{-1}$ or so. Our findings for $\alpha$ require that winds with low $\alpha$ must take on a REWS profile that lowers the turbine's equivalent wind speed below the hub-height wind speed (and vice versa for the high $\alpha$ case).

The surface-layer scaling parameter $\zeta$ efficiently segregates this turbine's power production into high and low cases. However, the $\zeta$ impacts on power are small, constrained to less than 20-kW (1.3% of rated) difference from the mean in either the high or low case (Fig. 14). Like $\alpha$, $\zeta$ varies strongly with both time of day (convective in local daytime hours and stable at night) and direction (stable when coming over complex terrain but convective otherwise), but $\alpha$ and $\zeta$ do not correlate linearly with each other by direction, further obfuscating attempts to draw stability conclusions from these metrics at this location.

eawe european academy of wind energy
Open Access WIND ENERGY SCIENCE DISCUSSIONS

The direct assessment of wind veering and backing, $\beta_{bulk}$, only shows small significant changes in power at wind speeds above 10 m s$^{-1}$ (Fig. 15). At those speeds, low $\beta_{bulk}$ (backing) leads to less power production than the mean case (under 10 kW or 0.7% of rated) while high $\beta_{bulk}$ cases (veering) leads to greater power production than the mean case (up to 20 kW or 1.3% of rated). These results agree with simulations (Wagner et al., 2010). However, at another (flat) site, Sanchez Gomez, and Lundquist (2019) found that both veer and backing decrease power compared to cases with no veering or backing; that study distinguished high veer from low veer, whereas we only contrast veering and backing. Like $\alpha, \beta_{bulk}$ lacks information about the inflow wind speeds. However, simply using REWS would mitigate this problem. $\beta_{bulk}$ shows a consistent daily cycle—all hours are dominated by backing at our site but backing weakens during the day (when $\alpha$ and $\zeta$ are convective) (Fig. 10e). $\beta_{bulk}$ does not show a strong directional cycle, except to say that westerly flow tends to be the only flow that introduces veer rather than backing and westerly flow is uncommon at this location (Fig. 9e). As with $\alpha$, care should be taken to consider the root cause of the directional sheer veer if it should be used by itself in future work. $\beta_{bulk}$ also suggests that $\beta_{total}$ is only a useful measurement at wind speeds less than 10 m s$^{-1}$, where the changes in power for veer and backing do not significantly differ from the mean (Fig. 17).

Overall, we find that REWS has the most predictive power for power production from an operational megawatt-scale wind turbine. REWS has the most significant results that occur over the largest portion of the power curve. In addition, because REWS simply functions as description of the wind at a given instant, rather than a prescription (such as stability that might be affected by factors such as topography), REWS is the simplest metric to understand and apply. Thus, findings for both high and low $\Delta\text{REWS}_L$ and $\Delta\text{REWS}_{N-NTF}$ likely hold at other locations and for other seasons and conditions, although the relationship between the frequency of occurrence of high and low cases would likely change at other locations.

Such results show that improvements in power production prediction in region II of a power curve are certainly greater on average than 15 kW (1% of rated power) for both high and low cases of $\Delta\text{REWS}_{N-NTF}$ or $\Delta\text{REWS}_L$ compared to the mean. The maximum increases in power production prediction can also exceed 4% of rated power, or even more when compared to the average power at a given wind speed. REWS is straightforward to implement and does not rely on assumptions or presumptions about the wind or stability.

The next step of this work would be to implement REWS into controls schemes for individual turbines or for entire wind farms. However, to do so, accurate measurements must be made of inflow across the rotor diameter from towers or remote sensing instruments. Likewise, for implementation into a wind farm's controls, these measurements would have to be spatially co-located somewhat with the turbine(s) they would affect, as inflow directions can change across the dimensions of a wind farm. Hub-mounted lidars are a promising method of such inflow characterization (Harris et al., 2007; Mikkelsen et al., 2013). Applying these methods to that inflow could help align the turbines further to maximize the potential of the inflow (Wagner et al., 2010; Fleming et al., 2014). Although this study found no meaningful difference between ΔREWS and

$\Delta \text{REWS}_{\theta}$, other locations with greater directional veer, influenced by meteorological phenomena such as cold pools (Wilczak et al., 2019; Redfern et al., 2019) or offshore decoupling (Bodini et al., 2019b), could find a more significant impact of the wind direction on the REWS.

**Code and data availability**

Currently, the data are not publicly available at the request of the wind farm owner/operator. The meteorological data may become available in the future at the DOE A2e data portal at https://a2e.energy.gov/about/dap.

**Author contributions**

JKL brought attention to this issue to PM during his senior undergraduate year at the University of Colorado Boulder for use as an independent study project. JKL and PM coordinated with PF to conduct analysis on the issue on a data set that PF, PM, and JKL were already using for other research. PM wrote the initial draft and created all figures; this draft was then reviewed, edited, and revised by PF and JKL. The final draft was made by PM based on suggested changes.

**Competing interests**

The authors declare that they have no conflict of interest.

**Disclaimer**

The views expressed in the article do not necessarily represent the views of the DOE or the U.S. Government.

**Financial support**

This analysis was supported by the National Science Foundation CAREER Award (AGS-1554055) to JKL.

This work was authored in part by the National Renewable Energy Laboratory, operated by Alliance for Sustainable Energy, LLC, for the U.S. Department of Energy (DOE) under Contract No. DE-AC36-08GO28308. Funding provided by the U.S. Department of Energy Office of Energy Efficiency and Renewable Energy Wind Energy Technologies Office. The views expressed in the article do not necessarily represent the views of the DOE or the U.S. Government. The U.S. Government retains and the publisher, by accepting the article for publication, acknowledges that the U.S. Government retains a nonexclusive, paid-up, irrevocable, worldwide license to publish or reproduce the published form of this work, or allow others to do so, for U.S. Government purposes.

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
