# Peer review of "How wind speed shear and directional veer affect the power production of a megawatt-scale operational wind turbine"

_Wind Energy Science, 2019_

## Referee Comment (RC1) · Melinda Marquis (Referee) · 3 Mar 2020

This paper quantifies the power production of a wind turbine in the high plains of North America to wind speed shear, directional veer, and atmospheric stability. Various metrics of each are analyzed. The log wind-shear exponent, bulk rotor-disk-layer veer, total rotor-disk-layer veer and rotor-equivalent wind speed (REWS) are considered, along with permutations of many. REWS includes directional veer. The REWS metric had the most impact on power production at this site. The log wind-shear exponent metric is not as useful and is sensitive to chosen critical values of it, probably at least partly because it represents not the actual wind speeds hitting the rotor disk but only

a quantity of wind shear. This study found only small impacts of directional veer and stability compared to shear on power production.

The authors explain how their findings align or conflict with previous studies, and provide possible explanations for such similarities and differences in findings.

Tables 1 and 2 provide key information in this paper. Addition of more space (empty lines) between rows in Table 1 would make it more readable.

The findings in paper are valuable because they can be used to estimate more accurate estimates of power production, which are important not only to energy system planners and grid operators, but also to financiers who facilitate development of wind plants.

---

## Referee Comment (RC2) · Anonymous Referee #2 · 5 Jun 2020

The paper discusses the effect of wind shear and veer as well as atmospheric stability on the power production in the partial load region of wind turbines. The analysis is based on real measurement data. The authors use different methods to describe the wind field situation across the turbine swept area. The description of the used data and data handling e.g. filtering is very detailed. It is interesting to see, that the usable data drops down to 30% of the available data and also reflects the difficulties to perform such investigations based on real data where the inflow conditions can not be controlled and repeated, respectively.

The paper is well written and easy to read even though there are many aspects touched

in this paper. The results are nicely summarised and put into context with other findings. I have no suggestions on how this paper could be improved — good work.

---

## Author Comment (AC1) · 30 Jun 2020

The reviewer's comment is in black, the author's comment is in red.

Thank you, Dr. Marquis, for your kind and thoughtful comments as well as the time spent working on this. We are appreciative that the findings are of interest to you and that you as well see the reach of interest of this work.

This paper quantifies the power production of a wind turbine in the high plains of North America to wind speed shear, directional veer, and atmospheric stability. Various metrics of each are analyzed. The log wind-shear exponent, bulk rotor-disk-layer veer, total rotor-disk-layer veer and rotor-equivalent wind speed (REWS) are considered, along with permutations of many. REWS includes directional veer. The REWS metric had the most impact on power production at this site. The log wind-shear exponent metric is not as useful and is sensitive to chosen critical values of it, probably at least partly because it represents not the actual wind speeds hitting the rotor disk but only a quantity of wind shear. This study found only small impacts of directional veer and stability compared to shear on power production.

The authors explain how their findings align or conflict with previous studies, and provide possible explanations for such similarities and differences in findings.

Tables 1 and 2 provide key information in this paper. Addition of more space (empty lines) between rows in Table 1 would make it more readable.

This has been fixed.

The findings in paper are valuable because they can be used to estimate more accurate estimates of power production, which are important not only to energy system planners and grid operators, but also to financiers who facilitate development of wind plants.

---

## Author Comment (AC2) · 30 Jun 2020

The reviewer's comment is in black, the author's comment is in red.

The authors appreciate the time spent reviewing and the thought put into the positive comments. Additionally, we are grateful that detail in sections (such as data filtering) was useful and of interest to the reviewer.

The paper discusses the effect of wind shear and veer as well as atmospheric stability on the power production in the partial load region of wind turbines. The analysis is based on real measurement data. The authors use different methods to describe the wind field situation across the turbine swept area. The description of the used data and data handling e.g. filtering is very detailed. It is interesting to see, that the usable data drops down to 30% of the available data and also reflects the difficulties to perform such investigations based on real data where the inflow conditions can not be controlled and repeated, respectively.

The paper is well written and easy to read even though there are many aspects touched in this paper. The results are nicely summarized and put into context with other findings. I have no suggestions on how this paper could be improved, good work.